# The Network for the Detection of Atmospheric Composition Change (NDACC): History, status and perspectives

Martine De Mazière[1], Anne M. Thompson[2], Michael J. Kurylo[3], Jeannette Wild[4], Germar Bernhard[5], Thomas Blumenstock[6], G. Braathen[7], James Hannigan[8], Jean-Christopher Lambert[1], Thierry Leblanc[9], Thomas J. McGee[2], Gerard Nedoluha[10], Irina Petropavlovskikh[11], Gunther Seckmeyer[12], Paul C. Simon[1], Wolfgang Steinbrecht[13], Susan Strahan[3]

[1]Royal Belgian Institute for Space Aeronomy (BIRA-IASB), Brussels, B-1180, Belgium
[2]NASA-Goddard Space Flight Center, Earth Sciences Division, Greenbelt, MD 20771, USA
[3]Universities Space Research Association, Goddard Earth Science, Technology and Research, NASA-Goddard Space Flight Center, Greenbelt, MD  20771-0001, USA
[4]Climate Prediction Center, NOAA Center for Weather and Climate Prediction, College Park, MD  20740,  USA
[5]Biospherical Instruments, Inc., San Diego, CA  92110-2621, USA
[6]Institute for Meteorology and Climate Research (IMK), Karlsruhe Institute of Technology (KIT), Karlsruhe, 76021, Germany
[7]Atmospheric Environment Research Division, Research Department, World Meteorological Organizqtion (WMO), Geneva, Switserland
[8]Atmospheric Chemistry Observations and Modeling, National Center for Atmospheric Research,  Boulder, CO  80305-5602, USA
[9]Table Mountain Facility, Jet Propulsion Laboratory,  Wrightwood, CA  92397-0367, USA
[10]Remote Sensing Division, Naval Research Laboratory, Washington, DC  20375-5351,USA
[11]NOAA Earth System Research Laboratory, Global Monitoring Division, Boulder, CO  80305-3328, USA
[12] Institute for Meteorology and Climatology, University of Hannover,  Hannover, D-30419, Germany
[13]Deutscher Wetterdienst, Meteorologisches Observatorium, Hohenpeissenberg, D-82383, Germany

*Correspondence to*: Martine De Mazière (martine.demaziere@aeronomie.be)

**Abstract.**

The Network for the Detection of Atmospheric Composition Change (NDACC) is an international global network of more than 90 stations making high quality measurements of atmospheric composition that began official operations in 1991 after five years of planning.  Apart from sonde measurements, all measurement in the Network are performed by remote sensing techniques. Originally named the Network for the Detection of Stratospheric Change (NDSC), the name of the Network was changed to NDACC in 2005 to better reflect the expanded scope of its measurements. The primary goal of NDACC is to establish long-term databases for detecting changes and trends in the chemical and physical state of the atmosphere (mesosphere, stratosphere and troposphere) and to assess the coupling of such changes with climate and air quality.  NDACC's origins, station locations, organizational structure and data archiving are described. NDACC is structured around categories of ground-based observational techniques (sonde, LIDAR, microwave radiometers, Fourier-transform Infrared, UV-visible DOAS-type and Dobson/Brewer spectrometers, as well as spectral UV radiometers), timely cross-cutting themes (ozone, water vapour, measurement strategies, cross-network data integration), satellite measurement systems, and theory and analyses. Participation in NDACC requires compliance with strict measurement and data protocols to ensure that the Network data are

of high and consistent quality. To widen its scope, NDACC has established formal collaborative agreements with eight other Cooperating Networks and GAW. A brief history is provided, major accomplishments of NDACC during its first 25 years of operation are reviewed, and a forward-looking perspective is presented.

## 1 Introduction

**1.1 Atmosphere Composition Issues in the 1970s and 1980s. Scoping an International Network**

When the scientific community looks back on the origins of research into measuring and understanding changes in global chemical composition, two phenomena are usually mentioned. One relates to regional air quality and the first characterization of photochemical "smog." Historians cite reports of threatening air quality as early as the 19th century but generally date studies of air pollution back to the 1950s when the chemical and physical processes leading to unhealthy urban environments

were first formulated. Second, during the 1960s and 1970s scientists began to consider chemical threats to the atmosphere as a whole. This was inspired by views of our planet from space and was given a boost from measurement projects that were initiated during the 1957-58 International Geophysical Year (IGY). During IGY, background ground-based monitoring stations began to measure in-situ surface concentrations of gases like carbon dioxide and methane, and total column ozone together with related constituents, many of which were heavily concentrated in the stratosphere. Unlike many short-lived

chemical pollutants, carbon dioxide and lower stratospheric ozone have long lifetimes and more uniform distributions globally. Furthermore, they are related to the radiative properties of the atmosphere. Water vapor and carbon dioxide are primary greenhouse gases and the thickness of the ozone column abundance determines the amount of ultraviolet (UV) radiation at the earth's surface.

Concerns about global ozone intensified with the realization that stratospheric ozone chemistry included catalytic cycles

involving reactive halogens, nitrogen and hydrogen (Bates and Nicolet, 1950; Crutzen, 1974; Stolarski and Cicerone, 1974; Molina and Rowland, 1974). Early spectroscopic balloon measurements confirmed the presence of trace species like $NO_2$, $HNO_3$, and HCl (Murcray et al., 1968; Murcray et al., 1973; Ackerman et al., 1974; Williams et al., 1976). With expanding space programs and proposals for large fleets of supersonic commercial aircraft, theoretical studies looked at possible threats to stratospheric ozone from rocket and aviation exhaust (Johnston, 1971). Scientists were also beginning to consider the

growing use of chlorofluorocarbons (CFCs) in myriad applications (Stolarski et al., 1974; Cicerone et al., 1974). Shortly thereafter, laboratory studies that measured the rates of free radical reactions, coupled with simple models, predicted global damage to ozone in the middle and upper stratosphere due to aviation exhaust emissions and to industrial halogenated compounds. Even the relatively inert $N_2O$, a byproduct of nitrogen fertilizers in wide use, would destroy ozone if it upset the natural balance of reactive nitrogen in the stratosphere.

Following a 1971 meeting of atmospheric scientists, the US "Stratospheric Protection Act of 1971," set up a Federal program of stratospheric research that was to report to the Congress within two years (Senate Congressional Record, September 21, 1971). In the fall of 1971, Congress assigned the US Department of Transportation (DOT) to conduct the two-year Climatic

Impact Assessment Program (CIAP), an international effort to assess the impact of climatic changes that might result from the introduction of propulsion effluents in the stratosphere (Grobecker, 1974). In 1972 the United Nations Conference on the Human Environment was held; its report (http://www.un-documents.net/aconf48-14r1.pdf) is considered a classic in the history of atmospheric chemistry.  As a result, programs like the French-UK COVOS (Comité d'Études sur les Conséquences

des Vols Stratosphériques) were initiated to assess the potential damage to future stratospheric ozone levels.  The US National Aeronautics and Space Administration (NASA) was given a long-term mandate in its FY1976 authorization bill to perform research concerned with the possible depletion of the ozone layer (covering all aspects of stratospheric chemistry, from laboratory investigations of chemical reaction rates, to ground-based and in-situ measurements of trace gases and computer modeling to simulate the present atmosphere and to predict the future). This mandate to perform research on the depletion of

the Earth's ozone layer soon led to the establishment of NASA's Upper Atmosphere Research Program (UARP). Numerous research projects supported by UARP complemented the first satellite measurements of global ozone by backscatter UV techniques (BUV) that started with the USSR Kosmos missions in 1964-1965 (Iozenas et al., 1969) and NASA's Orbiting Geophysical Observatory in 1967-1969 (Anderson et al., 1969) and BUV on Nimbus 4 in 1970-1975 (Heath et al., 1973).  The first European atmospheric research from space was based on solar occultation and limb emission instruments operated on the

Spacelab laboratory module, the latter built in cooperation between NASA and the European Space Research Organization (ESRO) that became the European Space Agency (ESA) in May 1975. After the pioneering flight of Spacelab 1 in 1983, 21 more Spacelab missions occurred between 1983 and 1998, among which three Atmospheric Laboratory for Application and Science (ATLAS) space shuttle missions in 1992-1994 (Miller et al., 1994). The ATLAS missions carried a set of instruments provided by the US and Europe, some in collaboration with Japan, and were occasionally complemented with additional

instruments operated on free-flying satellites launched from the space shuttle (ESA's EURECA and the German ASTRO-SPAS).  The US National Oceanic and Atmospheric Administration's (NOAA's) Upper Air Branch of the National Weather Service also began analyses of stratospheric measurements from ground-based and satellite data in the late 1970's.

The discovery of the Antarctic ozone hole (Farman et al., 1985) transformed atmospheric chemistry and made it clear that more detailed stratospheric observations were needed to help determine its origin.  Responding to predictions of ozone

depletion in the mid- to upper stratosphere (Molina and Rowland, 1974; Crutzen, 1974; Cadle et al., 1975), many countries had banned CFCs from certain applications in the 1970s.  Nevertheless, the morphology of Antarctic ozone loss and its severity were beyond any causal theories of the time.  Ground-based and airborne campaigns conducted in 1986 and 1987 provided evidence for a direct link between ozone depletion in the Antarctic stratosphere and catalytic halogen reaction cycles and indicated that the basic processes responsible for polar ozone loss involved heterogeneous reactions that took place on

atmospheric ice particles that formed at temperatures below the 185K potential temperature level.  Over the next decade, the results from subsequent aircraft campaigns in the Arctic demonstrated the vulnerability of the stratospheric ozone layer in both polar regions as well as at mid-latitudes. However, the mere observation of Antarctic ozone depletion in austral spring, together with considerable advances in the technology required to measure other stratospheric species from the ground, suggested that it was time to consider assembling a more detailed stratospheric monitoring program.  At that same time, the 1985 International

Vienna Convention for the Protection of the Ozone Layer gave a political mandate for comprehensive long-term monitoring of the ozone layer.  Thus, in March 1986 NASA, NOAA and the Chemical Manufacturers Association (CMA) convened a workshop in Boulder (Colorado, US) to ascertain the feasibility of developing a long-term observational network specifically designed to provide the earliest possible detection of changes in the composition and structure of the stratosphere and, more importantly, the means to understand the causes of those changes.  Measurement priorities and goals were defined, station placements were considered, and potential instrumentation was evaluated.  Many instruments were under development (ozone lidar (light detection and ranging)), some demonstrated (microwave radiometry for $H_2O$, UV-Visible spectrometry for $NO_2$, Fourier-transform infrared spectrometry (FTIR) for HCl), and some proposed (microwave for $N_2O$, FTIR for several more species).  However, only the Dobson ozone spectrometer was fully operational at that time.  At a 1989 meeting in Geneva, NASA, NOAA and the World Meteorological Organization (WMO) convened a forum at which several international agencies and institutions participated.  At that meeting the actual organizational structure of the NDSC was formalized (Kurylo and Solomon, 1990). Annual Steering Committee meetings for the Network commenced in 1990; in 1991 and, after five years of planning, the NDSC began official operations. These international planning meetings (**Table 1)** had led to the realization that such a research and monitoring program needed to be global.  Thus, NDSC represented from its beginning a consortium of countries and sponsoring organizations, with endorsement from the United Nations Environment Programme (UNEP), WMO, and the International Ozone Commission ($IO_3C$), a body of the IUGG/IAMAS (International Union of Geodesy and Geophysics/International Association of Meteorology and Atmospheric Science).

The stratospheric ozone focus was an obvious integrating theme during the early NDSC years.  However, the scope of measurement requirements was broadened by regular collaboration with field measurement programs and experiments, interdisciplinary data analysis and modeling activities and assessments.  Therefore, in 2005, NDSC changed its name to NDACC, Network for the Detection of Atmospheric Composition Change, to reflect the broadened scope. Many trace gases measured in NDACC and by its partners (see Sect. 2.2) are as important to climate issues and/or air quality as they are to ozone depletion, as recognized in today's NDACC objectives, listed as follows:

- establish long-term databases for detecting changes and trends in atmospheric composition, and understand their impacts on the mesosphere, stratosphere and troposphere;
- establish scientific links and feedbacks between changes in atmospheric composition, climate and air quality;
- validate atmospheric measurements from other platforms (i.e., satellites, aircraft and ground-based);
- provide critical datasets to help fill gaps in satellite observations;
- provide collaborative support to scientific field campaigns and to other chemistry and climate-observing networks; and
- provide validation and development support for atmospheric models.

Hereafter we will use the current acronym NDACC whenever we refer to the Network.

## 1.2 Structure of overview paper

NDACC has provided a unique, enduring framework for the international community to make long-term ground-based measurements of atmospheric composition on a global scale. To celebrate the 10[th] and 20[th] anniversaries of NDACC, symposia highlighting the network scientific achievements were held in 1991 (Arachon, France) and Réunion Island (2011) respectively. For the 25[th] anniversary NDACC decided to publish a feature article in The Earth Observer (Kurylo et al., 2016) and to assemble an inter-journal Special Issue in the journals *Atmospheric Chemistry and Physics, Atmospheric Measurement Techniques*, and *Earth System Science Data*. This paper is the introductory paper for this Special Issue. The organizational structure and workings of NDACC, remarkably adaptable and important to its success, are described in Section 2. Highlights of scientific accomplishments of NDACC over the past 25 years appear in Section 3. Section 4 anticipates further developments in network configurations and Section 5 is a perspective on the future of NDACC as we look at current issues in global atmospheric composition and dynamics.

**Table 1. NDSC/NDACC Meeting History & Key Actions and Steering Committee (SC) Chairs and Co-Chair Elections**

| Year | Location | Actions |
|------|----------|---------|
| 1986 | Boulder, CO, USA | Concept and feasibility of the network evaluated |
| 1989 | Geneva, Switzerland | Managerial and organizational structure of the NDSC formalized; SC Chair (Michael J. Kurylo) and Vice-Chair (R. A "Tony" Cox) elected |
| 1990 | Washington, DC, USA | 1st annual NDSC SC Meeting; 5 Primary Stations designated covering both hemispheres |
| 1991 | Abingdon, UK | 2nd annual SC Meeting; official network operations began endorsed by UNEP, WMO, and IO3C; complementary measurement opportunities discussed; data protocol finalized; official NDSC Data Host Facility (DHF) established at NOAA with mirroring at the British Atmospheric Data Center (BADC) and at the Norwegian Institute for Air Research (NILU) |
| 1992 | Paris, France | 3rd annual SC Meeting; evaluation of new instruments; Complementary Sites designated; NDSC Data Host Facility (DHF) begins archiving data from multiple sites |
| 1993 | Wrightwood, CA US | 4th annual SC Meeting; Instrument Validation Policy document finalized; new Complementary Sites approved; potential Theory and Analysis investigators identified; protocol for SC Elections and Appointments finalized; Mike Kurylo re-elected to 3-year term as SC Chair |
| 1994 | Queenstown, NZ | 5th annual SC Meeting; Protocol for Instrument Intercomparisons finalized; Spectral UV measurements added to network |
| 1995 | Leuven, Belgium | 6th annual SC meeting; additional Complementary Measurement activities approved; NDSC web site announced; Instrument-Specific Validation Appendices added to Validation Protocol; Rudy Zander elected as SC Vice-Chair, replacing Tony Cox, who had resigned; Mike Kurylo re-elected as SC Chair; |
| 1996 | Waikoloa, HI, USA | 7th annual SC Meeting; formal presentations by Instrument Working Groups representing the various NDSC-designated instrument types and by the Satellite and the Theory and Analysis Working Groups; Mike Kurylo re-elected to 3-year term as SC Chair |
| 1997 | Spitsbergen, Norway | 8th annual SC Meeting; status of and plans for NDSC mobile instrument reviewed; Dobson/Brewer Instrument Working Group added |
| 1998 | Réunion Island, France | 9th annual SC Meeting; endorsement given to develop a new observatory site at Maïdo; Rudy Zander re-elected to 3-year term as SC Vice-Chair |
| 1999 | Sapporo, Japan | 10th annual SC Meeting; designations of Primary and Alternate Working Group Representatives changed to Co-Representatives; SC Chair and Vice-Chair positions re-designated as Co-Chairs; Mike Kurylo elected to 3-year term as SC Co-Chair; Rudy Zander's position changed from SC Vice-Chair to SC Co-Chair; new Ex-Officio positions established on the SC |
| 2000 | Thun, Switzerland | 11th annual SC Meeting; annual station report forms standardized; 10 year NDSC Anniversary Symposium to be scheduled in 2001 |
| 2001 | Arcachon, France | 12th annual SC Meeting held in conjunction with an International Symposium celebrating 10 years of NDSC operations; Rudy Zander re-elected as SC Co-Chair |
| 2002 | Toronto, Canada | 13th annual SC Meeting; special NDSC session to be conducted at the 2003 joint EGS/AGU meeting; draft of first NDSC Newsletter presented; Rudy Zander resigned as NDSC Co-Chair due to his University retirement; Paul Simon elected to serve the remaining 2 years of Rudy Zander's term; Mike Kurylo re-elected to 3-year term as SC Co-Chair |

| | | | |
|---|---|---|---|
| | 2003 | Wellington, NZ | 14th annual SC Meeting; final version of NDSC Newsletter presented; creation of an NDSC leaflet discussed |
| | 2004 | Andøya, Norway | 15th annual SC Meeting; discussions on how to make NDSC connections to global change and the troposphere more visible; with the expiration of Paul Simon's position as SC Co-Chair, he was named to an Ex-Officio position on the SC and Geir Braathen was elected as the new SC Co-Chair |
| 5 | 2005 | Tenerife, Spain | 16th annual SC Meeting; Mike Kurylo was re-elected as SC Co-Chair; the name of NDSC changed to the Network for the Detection of Atmospheric Composition Change (NDACC), to better reflect the expanded focus of its measurements |
| | 2006 | OHP, France | 17th annual SC Meeting; report on water vapor measurement techniques presented; options for a new NDACC logo discussed |
| 10 | 2007 | Waikoloa, HI, USA | 18th annual SC Meeting; discussions on how to make external network affiliations more meaningful; Geir Braathen re-elected as SC Co-Chair |
| | 2008 | Kangerlussuaq and Ilulissat, Greenland | 19th annual SC Meeting; Primary and Complementary Site/Station designations replaced by NDACC-approved Measurement Site/Station; NDACC Cooperating Network affiliation established; tropospheric water vapor lidars and water vapor sondes approved as NDACC techniques; Mike Kurylo re-elected as SC Co-Chair. |
| 15 | 2009 | Geneva, Switzerland | 20th annual SC Meeting; 5 initial Cooperating Network affiliations approved |
| | 2010 | Queenstown, NZ | 21st annual SC Meeting; 6th Cooperating Network added; Symposium planned to commemorate 20 years of NDSC/NDACC Observations; Geir Braathen accepted re-election as SC Co-Chair |
| | 2011 | Réunion Island, France | 22nd annual SC Meeting held at NDSC/NDACC 20 years Anniversary symposium; 7th Cooperating Network added; rapid delivery data added to the DHF; Stuart McDermid elected as SC Co-Chair, replacing Mike Kurylo who stepped down |
| 20 | 2012 | Garmisch, Germany | 23rd annual SC Meeting; opening of new Maïdo Observatory on Réunion Island announced; Water Vapor Working Group announced publication of ISSI Scientific Report on Monitoring Atmospheric Water Vapour; role of NDACC measurements in the SPARC/IO3C/IGACO/NDACC (SI2N) Initiative on Past Changes in the Vertical Distribution of Ozone highlighted |
| | 2013 | Frascati, Italy | 24th annual SC Meeting; Martine De Mazière elected as SC Co-Chair, replacing Geir Braathen who stepped down |
| 25 | 2014 | Brussels, Belgium | 25th annual SC Meeting; Anne Thompson elected as SC Co-Chair, replacing Stuart McDermid who stepped down |
| | 2015 | La Jolla, CA, USA | 26th annual SC Meeting; Theory and Analysis Working Group announced the availability of model support files for several instrument types |
| | 2016 | Bremen, Germany | 27th annual SC Meeting; 25 years of successful NDSC/NDACC measurements and analyses highlighted in a feature article in NASA's *Earth Observer* newsletter; M. De Mazière re-elected for a second term as co-chair. |
| 30 | 2017 | Boulder, CO, USA | 28th annual SC meeting; A.M. Thompson re-elected for a second term as co-chair. |

## 2 The Organization and Workings of NDACC

### 2.1 Scope of Measurements, Stations and Objectives

Figure 1 illustrates the major atmospheric variables (constituents and physical parameters) that are measured within NDACC in order to achieve the full scope of network objectives listed in Section 1. Included are column and vertical profile
measurements that provide complementarity to satellite measurements of the same variable. For example, UV-visible DOAS-type instruments, that are shown along with Brewer and Dobson spectrometers, have kept myriad ozone satellite instruments calibrated and cross-calibrated since the start of NDACC and even two decades before.

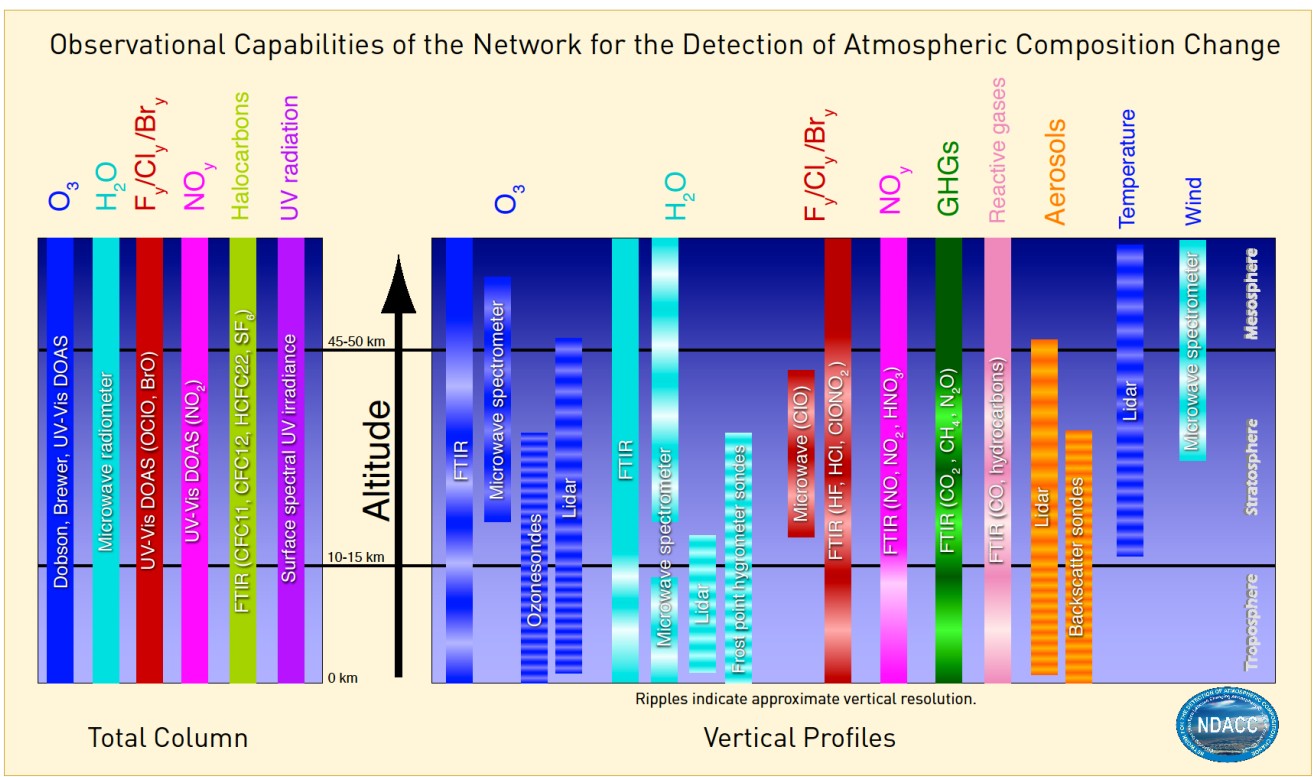

**Figure 1: NDACC measurement capabilities, including species and parameters measured, instrumental measurement techniques, and each measurements' approximate vertical resolution (indicated by the ripple).**

The right side of Fig. 1 depicts the vertical resolution of NDACC techniques used to measure various constituents throughout the troposphere and stratosphere up to the lower mesosphere. Note the use of microwave, lidar and FTIR along with balloon-borne soundings that, in many locations, offer the greatest vertical resolution up to the middle stratosphere.
A few more species are measured, e.g., HCFC-142b at some FTIR stations, and this list of species keeps increasing with time, conform the evolution of the measurement techniques, of the research interests and the changing abundances of some species.

Very soon, the UV-Visible MAXDOAS (Multi-AXis DOAS) technique will be included in the list of certified NDACC measurement techniques, enabling the observation of additional species like glyoxal as well as the observation of lower tropospheric profiles of $NO_2$, HCHO and $O_3$ (Kreher et al., 2018).

The full list of species for which data are available in the NDACC database are given in Box 1; for some species not listed in
Box 1, data are available upon request from the individual PIs.

Box 1. Full list of variables (in alphabetical order) for which data are available in the NDACC database at the moment of this publication.

| |
|---|
| Aerosol, BrO, $C_2H_2$, $C_2H_4$, $C_2H_6$, $CCl_2F_2$, $CCl_3F$, $CH_3OH$, $CH_4$, $CHF_2Cl$, Chlorine, $ClONO_2$, CO, $CO_2$, $COF_2$, $H_2CO$, $H_2O$ and Isotopologues, HCHO, HCl, HCN, HCOOH, HF, $HNO_3$, $N_2O$, $NH_3$, NO, $NO_2$, OClO, OCS, Ozone, $SF_6$, Temperature, Tropospheric Ozone, spectral UV irradiance, Wind |

An essential element of NDACC is the rigor of the measurements and their analyses, which since the first days of NDACC have been ensured by regular instrument and algorithm validation and intercomparison campaigns. A key ingredient of NDACC has been the establishment of written protocols detailing validation procedures, expectation of instrument and measurement quality standards and data analysis and reporting standards (http://www.ndaccdemo.org/data-documents/protocols/). This quality assurance lends considerable credence to the ground-based record which NDACC has
contributed to all the quadrennial WMO Scientific Assessments of Ozone Depletion (1991 to present).

The current map with certified active NDACC stations appears in Fig. 2. The 1986 Workshop envisioned an initial network structure of 6 primary stations, most of which would consist of several sites and host several instruments. An additional site at Table Mountain (California) was to be a 'test site' and, thus, became the first complementary site in the network. The established stations at Observatoire de Haute Provence (OHP), France, and Jungfraujoch, Switzerland, and at Mauna Loa,
Hawaii, were identified as principal contributors to the primary stations (Alpine Station and Hawaii Station, respectively). In 1991, 5 Primary Stations were actually established: Arctic, Alpine, Hawaii, Lauder New Zealand, and Antarctic Station. It was anticipated that numerous Complementary Stations, at which a smaller number of Network-approved instruments were in operation, or at which the measurement commitment was for a shorter period of time, would augment these Primary Stations. In 2008 the NDACC Steering Committee decided to remove the "Primary" and "Complementary" designations of NDACC
measurement sites / stations since their use was leading to some confusion and occasional misunderstanding. For example, some Complementary Stations had built up suites of instruments that were more comprehensive than those at some of the Primary Stations and many Complementary Stations had measurement commitments that were just as long-term as those at Primary Stations. Further, the designations had occasionally been misinterpreted to imply that the measurements at and data from Complementary Stations were of lesser quality than those at Primary Stations, whereas the requirements to become
affiliated with NDACC were identical for the two categories.

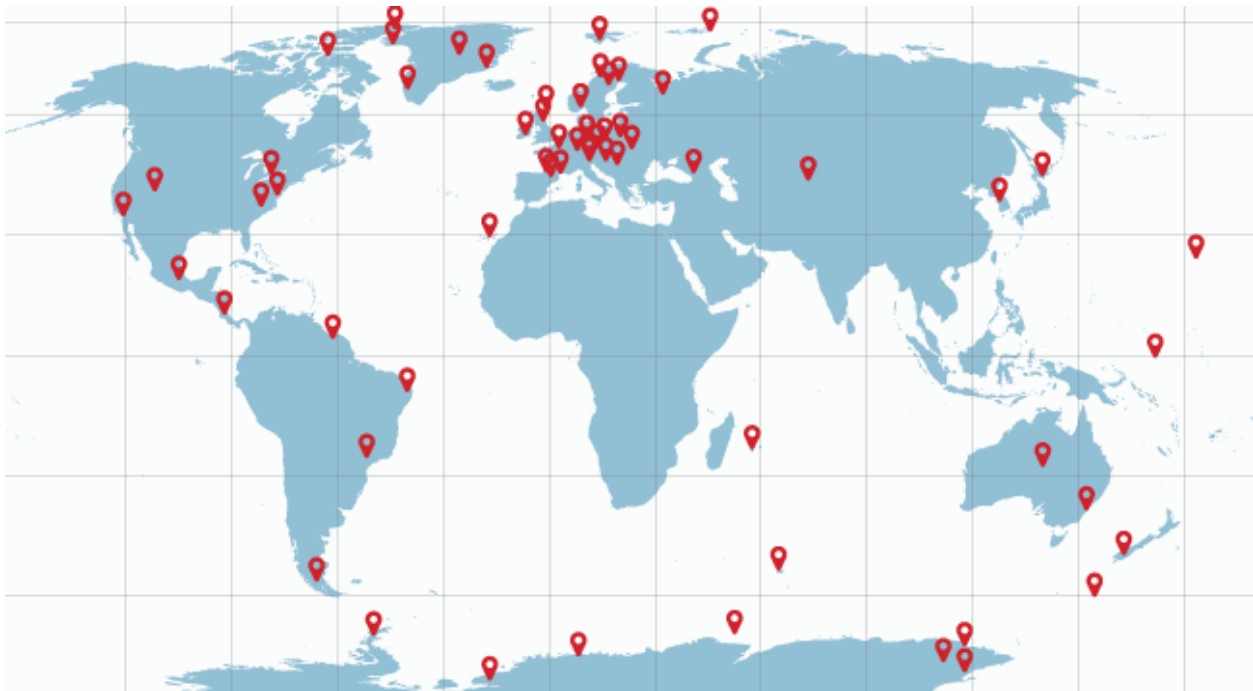

**Figure 2: Map of currently active NDACC Stations.**

In addition to the stations in Fig. 2, some new stations (in China, Africa, South America, …) have applied for affiliation to NDACC. While they are in the process of certification during which the compliance with the NDACC Protocols is verified,
they have the status of 'candidate stations' and do not appear on the map of active stations in Fig. 2.

## 2.2 Network Structure and Workings

Figure 3 illustrates the organizational structure of NDACC. Its basic structure is unchanged over 25 years but the details of Steering Committee (SC) composition, Working and Theme Groups, as well as partners and the Data Handling Facility (DHF, Sect. 2.3) have evolved over time. The present Steering Committee is led by two Co-Chairs; Table 1 lists the Co-Chairs who
have served since 1991 together with a history of SC Meetings and some ensuing actions. There are seven permanent Instrument Working Groups (Fig. 3) in NDACC, organized around instrument types: Dobson and Brewer, FTIR, lidar, microwave, sondes, spectral UV, UV-visible or UVVIS spectrometers. Two additional permanent Working Groups, on Satellites and on Theory and Analysis, contribute cross-cutting activities connected to multiple instrument types and liaise with data user communities. Two representatives from each of the nine Working Groups are members of the NDACC SC.
The Steering Committee also identified the need for four Theme Groups whose activities are typically of more limited duration and are organized around specific foci that may be relevant to several Instrument Working Groups. Therefore, the Theme Groups often involve participation from representatives from several Instrument Working Groups, and benefit from synergies between measurement techniques to address their foci. For example, a Water Vapor Theme group was established in 2006 to

assess the accuracy of various water vapor measurement techniques and resulted in an ISSI (International Space Science Institute) publication (Kämpfer, 2013). More recently the Water Vapor Theme Group has been developing a network-wide measurement strategy for atmospheric water vapor. Although its initial instrument orientation focused on frost point sondes, the strategy will coordinate all current NDACC water vapor measurements (e.g., lidar, microwave, FTIR).

Since its beginning, NDACC recognized the importance of new measurement capabilities and of existing capabilities whose heritage was developed external to the network as well as the strong scientific benefits of fostering collaborative measurement, analysis and quality assurance activities with other networks that were operating independently of NDACC and were collecting high-quality data. Accordingly, Cooperating Network affiliations with NDACC were initiated (Table 1) with data-sharing

10 protocols. In 2009 the first five Cooperating Networks were formalized. There are eight Cooperating Networks at this time (Table 2) and a representative of each one serves on the NDACC SC.

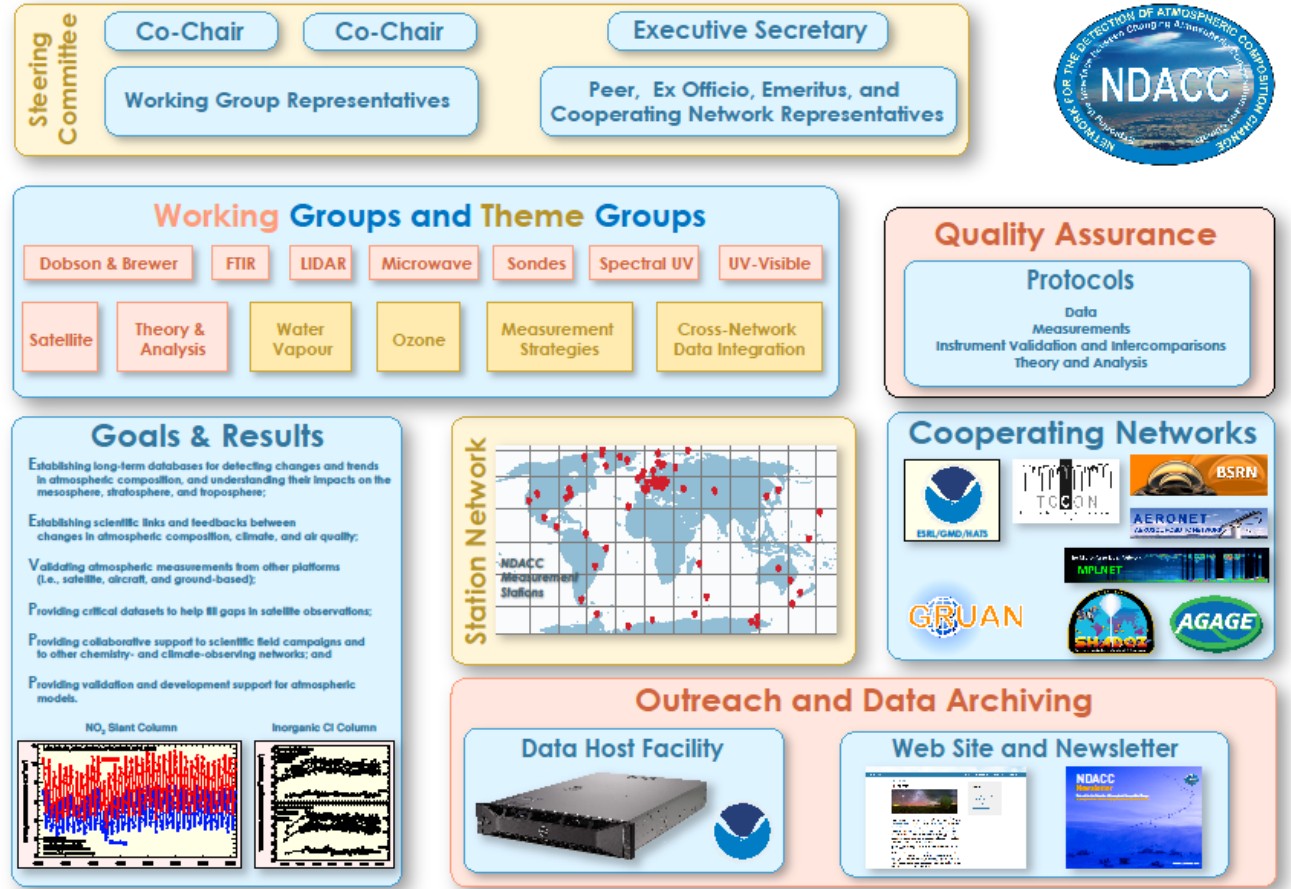

**Figure 3: Organizational structure of NDACC and the logo. More details of Steering Committee and Working and Theme Groups**

15 **composition, documents related to Group activities, along with the data, are found at the website http://www.ndacc.org.**

**Table 2. List of Cooperating networks**

| Cooperating Network | Website |
|---|---|
| AErosol RObotic NETwork (AERONET) – 2009 | http://aeronet.gsfc.nasa.gov |
| Advanced Global Atmospheric Gases Experiment (AGAGE) - 2009 | http://agage.eas.gatech.edu/index.htm |
| The Baseline Surface Radiation Network (BSRN) – 2011 | http://www.bsrn.awi.de |
| GCOS Reference Upper-Air Network (GRUAN) – 2013 | http://www.gruan.org |
| The Halocarbons and other Trace Species (HATS) – 2009 | http://www.esrl.noaa.gov/gmd/hats |
| The NASA Micro Pulse Lidar Network (MPLNET) – 2009 | http://mplnet.gsfc.nasa.gov |
| Southern Hemisphere Additional Ozonesondes (SHADOZ) - 2009 | http://tropo.gsfc.nasa.gov/shadoz |
| Total Carbon Column Observing Network (TCCON) - 2011 | http://www.tccon.caltech.edu |

Products useful for quantifying the feedbacks between climate change and atmospheric composition will require careful integration of information from sondes with instruments that supply integrated column values or low-resolution vertical profiles. In this effort, NDACC may expand its relationship with the sounding-focused GCOS (Global Climate Observing System) Reference Upper Air Network (GRUAN).

**2.3 Data Handling Facility (DHF)**

Initially the data from the 5 original primary and complementary stations were housed on a VAX/VMS system with access solely to the NDACC data providers. Mirrors of the NDACC database were housed at the British Atmospheric Data Centre (BADC) and the Norwegian Institute for Air Research (NILU) to provide offsite backup and distributed data access for international partners. The file format chosen in collaboration with UARS, EASOE (European Arctic Stratospheric Ozone Experiment) (European Commission, 1997) and other international projects was the simple ASCII Ames (Gaines and Hipskind, 1998) format. After the two-year validation period and internal publication, data were transferred to a public ftp site, and to database partners. In 2001 the satellite community asked the NDACC to consider use of the HDF format. The DHF managers have participated in the Generic Earth Observation Metadata Standard (GEOMS) initiative to develop reporting standards for calibration/validation data (De Mazière et al., 2002; https://avdc.gsfc.nasa.gov/index.php?site=1178067684). Today the NDACC DHF is based on dual Linux servers with dynamic failover, and houses data from 148 active instruments at 80 sites, as well as campaign data and data from past instruments. These data are publically available at ftp.cpc.ncep.noaa.gov/ndacc/station. The NDACC DHF engages in collaboration with AVDC (AURA Validation Data

Center), GAWSIS (Global Atmosphere Watch Station Information System), GECA (Generic Environment for Calibration/validation Analysis, Meijer et al., 2009), and WOUDC (World Ozone and Ultraviolet Radiation Data Centre, http://woudc.org/) where the NDACC database can be searched remotely; in some cases additional visualization tools are provided.

In an effort to provide a clearer and more direct path to access NDACC data, the NDACC web page has been redesigned. Data search tools include dynamic search by maps, instrument type, and station listing. The Content Management System (CMS) based design provides simple tools for updating documentation, and enforcing documentation requirements resulting in information that is more visible and more easily accessed than in the past. Figure 4 shows the newly designed web site which is available at www.ndacc.org.

**3. Important NDACC achievements during 25 Years of monitoring atmospheric composition change**

The following contributions of NDACC demonstrate the centrality of its measurements program and the invaluable roles played by consistent, standardized, long-term measurements that are organized in a network. These examples also illustrate how NDACC is integrated into other atmospheric activities like SPARC (Stratosphere-troposphere Processes and their Role in Climate), GCOS (Global Climate Observing System), IGACO (Integrated Global Atmospheric Chemistry Observations,

the $IO_3C$, CAMS (Copernicus Atmospheric Monitoring Service), C3S (Copernicus Climate Change Service), and WMO/GAW.

The most recent achievements are reported in the scientific papers in this Special Issue and listed and categorized in Annex A, or they are referenced in the below sections.

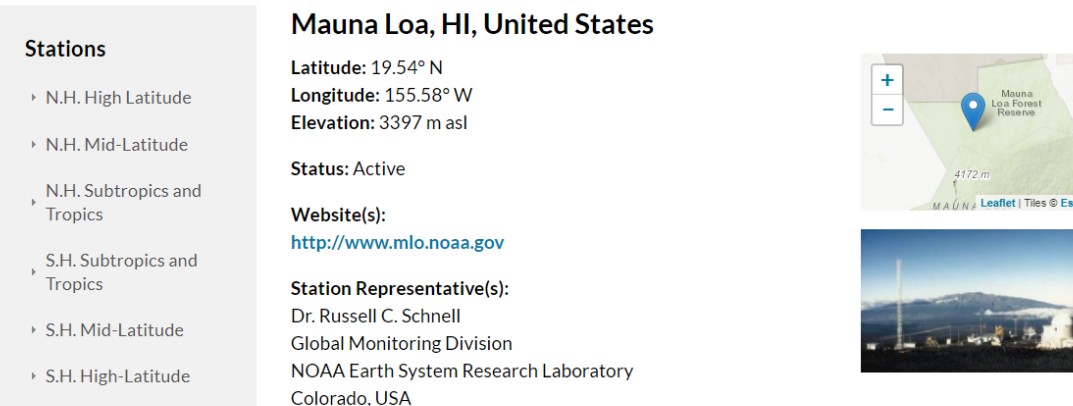

**Figure 4: The newly redesigned NDACC Web site at www.ndacc.org. Top: Stations are searchable by a dynamic map with filters. Bottom: Station pages allow direct access to data and metadata instrument description files.**

## 3.1 Long-term ozone monitoring

Figure 5 shows the blending of satellite data (SAGE, OSIRIS, ESA-CCI (Climate Change Initiative, http://cci.esa.int/), SWOOSH (The Stratospheric Water and OzOne Satellite Homogenized; https://www.esrl.noaa.gov/csd/groups/csd8/swoosh/) and GOZCARDS (Global OZone Chemistry And Related trace gas Data records for the Stratosphere, https://gozcards.jpl.nasa.gov/info.php)) merged datasets, SBUV-MOD (SBUV Merged Ozone dataset) with reference datasets from more than two decades of ozone measurements by NDACC lidars and microwave radiometers. Note that the QBO and the solar cycle, indicators of natural phenomena that affect stratospheric ozone amounts, are also shown. Ground-based microwave instruments provide continuous measurements of ozone, and can thus provide information on diurnal variations in ozone (Haefele et al., 2008; Parrish et al., 2014). This is of particular value for helping to interpret satellite measurements, which drift in local time.

The response of stratospheric ozone to the declining atmospheric abundance of ozone-depleting substances (ODS, represented by ESC "effective stratospheric chlorine" in the lower part of Fig. 5) is illustrated. The decline in ESC is a direct result of the Montreal Protocol (1987) and its Amendments and Adjustments. A decline in ozone can be seen at all sites until approximately 2000 when the onset of increases becomes apparent at the northern mid-latitude stations. Complete ozone recovery (i.e., to 1980 benchmark levels) from the effects halogen catalyzed destruction is projected to occur by mid-21st century at mid-latitudes and over the Arctic, and somewhat later for the Antarctic ozone hole. However, increases in greenhouse gas abundances over this same period are expected to cause concurrent stratospheric ozone increases.

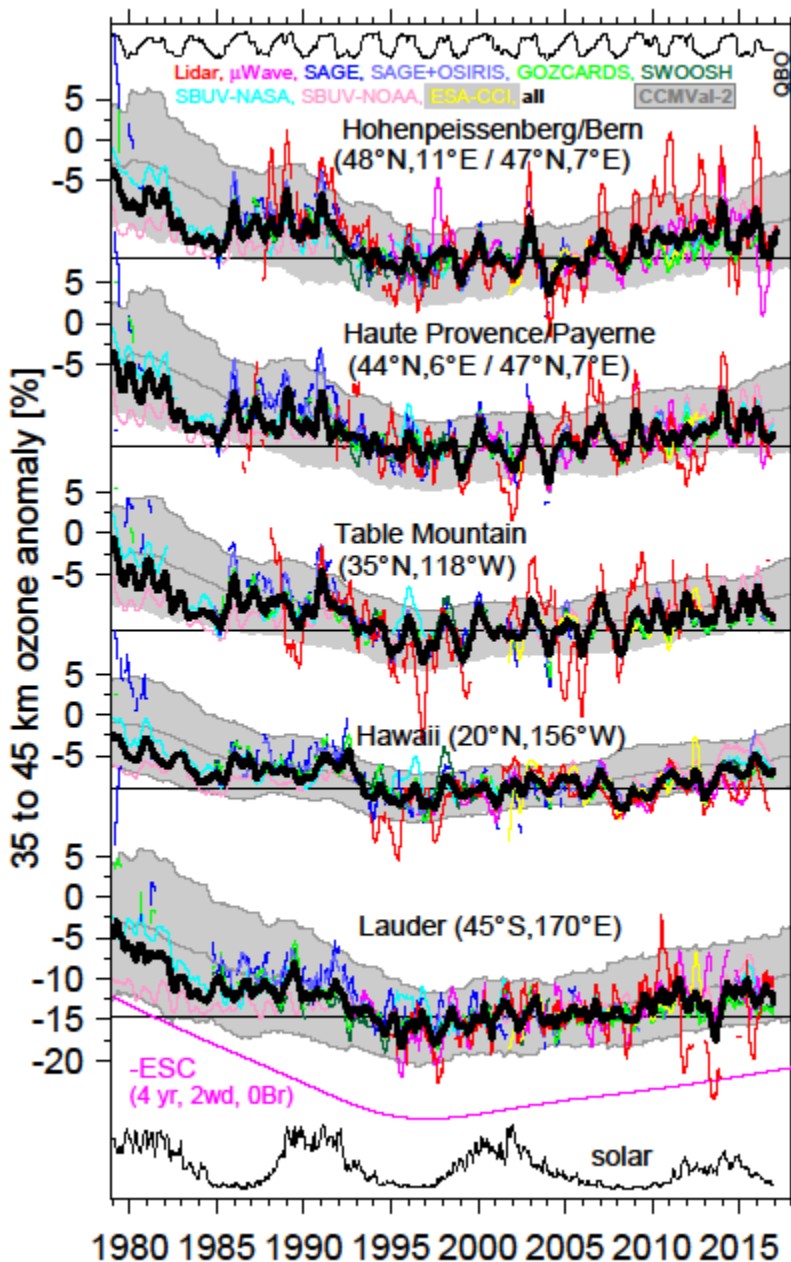

**Figure 5: The essential synergism of NDACC measurements with data from satellites. The grey area shows the range of model evaluations from the CCMVal-2 initiative. Also shown at the top/bottom are the expected natural variability (QBO / solar cycle) and the evolution of ESC (effective stratospheric chlorine). The data are used for verifying successful implementation of the Montreal Protocol.**

### 3.2 Constraining uncertainties in ozone absorption cross-sections

NDACC instrument scientists have been important participants in the Absorption Cross-Sections of Ozone (ACSO) activity conducted as a joint initiative of the IO$_3$C, WMO, and the IGACO O$_3$/UV subgroup to study, evaluate, and recommend the most suitable ozone absorption cross-section laboratory data to be used in atmospheric ozone measurements. Comparisons of NDACC ozone products generated by various ~~different~~ instrument types helped determine the range of uncertainty associated with the stratospheric temperature dependence of the instrument-specific absorption cross-sections that are operationally used in derivation of these data products. These determinations led to ACSO recommendations for using various spectroscopic data published in the literature and to conduct further laboratory measurements <http://www.wmo.int/pages/prog/arep/gaw/documents/FINAL_GAW_218.pdf >.

This activity supports the data analysis from the Dobson and Brewer networks in NDACC. In 2016, the IO$_3$C recommended replacing the Bass and Paur (1985) ozone cross-sections with those of Gorshelev et al. (2014) and Serdyuchenko et al. (2014), partly because use of the latter improved total ozone agreement between Dobson and Brewer instruments (Redondas et al., 2014). Koukouli et al. (2016) showed, in addition, the importance of correcting effective temperature errors in the Dobson spectrophotometers. Note that updates of Dobson data have been on-going. NOAA coordinates the data collection for 14 of these instruments, with seven of them (Mauna Loa, Boulder, Lauder, OHP, American Samoa, Wallops, South Pole) reporting to NDACC. A recent software update resulted in a re-evaluation of the Dobson ozone record for the NOAA instrument complement. The new records were compared to the original NDACC and WOUDC records (Evans et al., 2017). At the completion of the evaluation, new datasets were archived at NDACC and WOUDC. Another large re-processing effort that will update the NDACC Dobson and Brewer records with new absorption cross sections is underway.

### 3.3 Reference measurements for satellite validation

In the 1980s a few ozone monitoring stations - mainly equipped with Dobsons, Brewers, DOAS UV-visible spectrometers, lidars and ozonesondes - had already been used as ground-based references for the geophysical validation of TOMS column and SAGE-II and SBUV/2 profile data. With the advent of new generation satellite measurements in the 1990s (e.g., UARS, ATLAS, GOME, ADEOS, EOS-Terra) and 2000s (Odin, Envisat, SCISAT-1 ACE, EOS-Aura, MetOp, GOSAT), these pioneering validation activities have progressively developed to encompass all types of NDACC instruments and their complete portfolio of species and parameters. Validations based on single instruments at single stations have expanded to more comprehensive assessments using the network as a whole. The portfolio of NDACC data products has gradually been enhanced to meet emerging needs of the satellite community. To date, NDACC has contributed sustained support to the geophysical validation and algorithm evolution of over 50 space-based sounders. These include the series of nadir-viewing UV-visible and infrared, and the limb and occultation profilers on the UARS, Odin, Envisat, SCISAT-1 and EOS-Aura platforms (Figs. 6a and 6b). NDACC data have also been used to assess the stability and mutual consistency of multiple satellite data records

across a multi-decadal period, e.g., McPeters and Labow, 1996; McPeters et al., 2008; Fioletov et al., 2008; Antón et al., 2009; Flynn et al., 2014; Bak et al., 2015;

Koukouli et al., 2015; Hubert et al., 2016). NDACC is also supporting current operational missions like MetOp, Suomi-NPP, SAGE III/ISS, Sentinel-5p TROPOMI and JPSS-1, and is recognized as a key source of Fiducial Reference Measurements

5 (FRM) for the validation of the upcoming Copernicus atmospheric Sentinels 4 and 5 and the China High-resolution Earth Observation System (CHEOS).

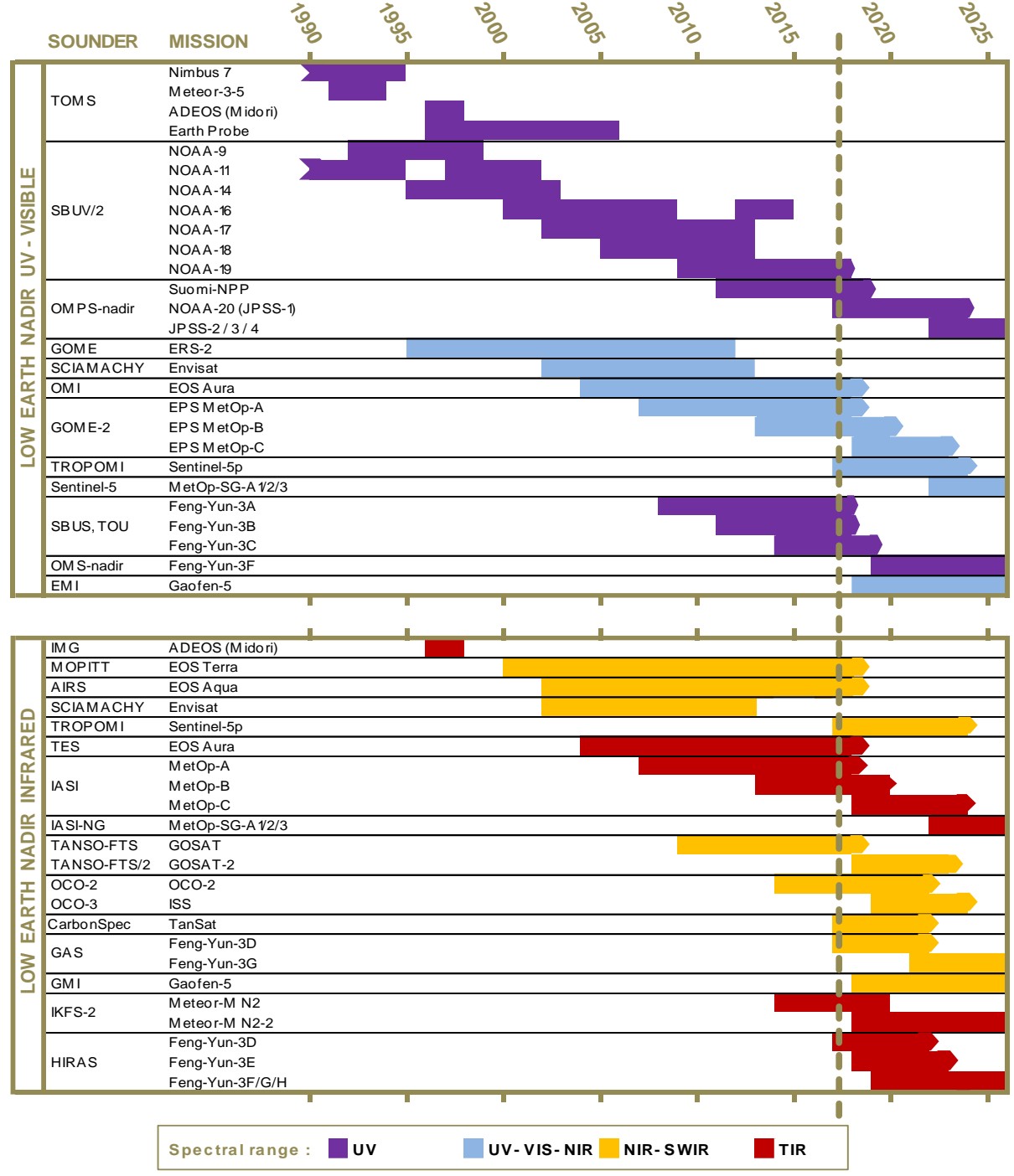

**Figure 6a: Low Earth Orbit nadir-viewing satellite sounders supported by NDACC, from the inception of the NDSC through present day and beyond. Upper chart: backscatter UV (O₃) and UV / VIS (O₃, NO₂, BrO, HCHO…) Lower chart: backscatter NIR / SWIR (typically CH₄, CO, H₂O, N₂O) and MIR / TIR emission (O₃, CH₄, CO, N₂O, H₂O, HNO₃, HCl, CFC-11, CFC-12…)**

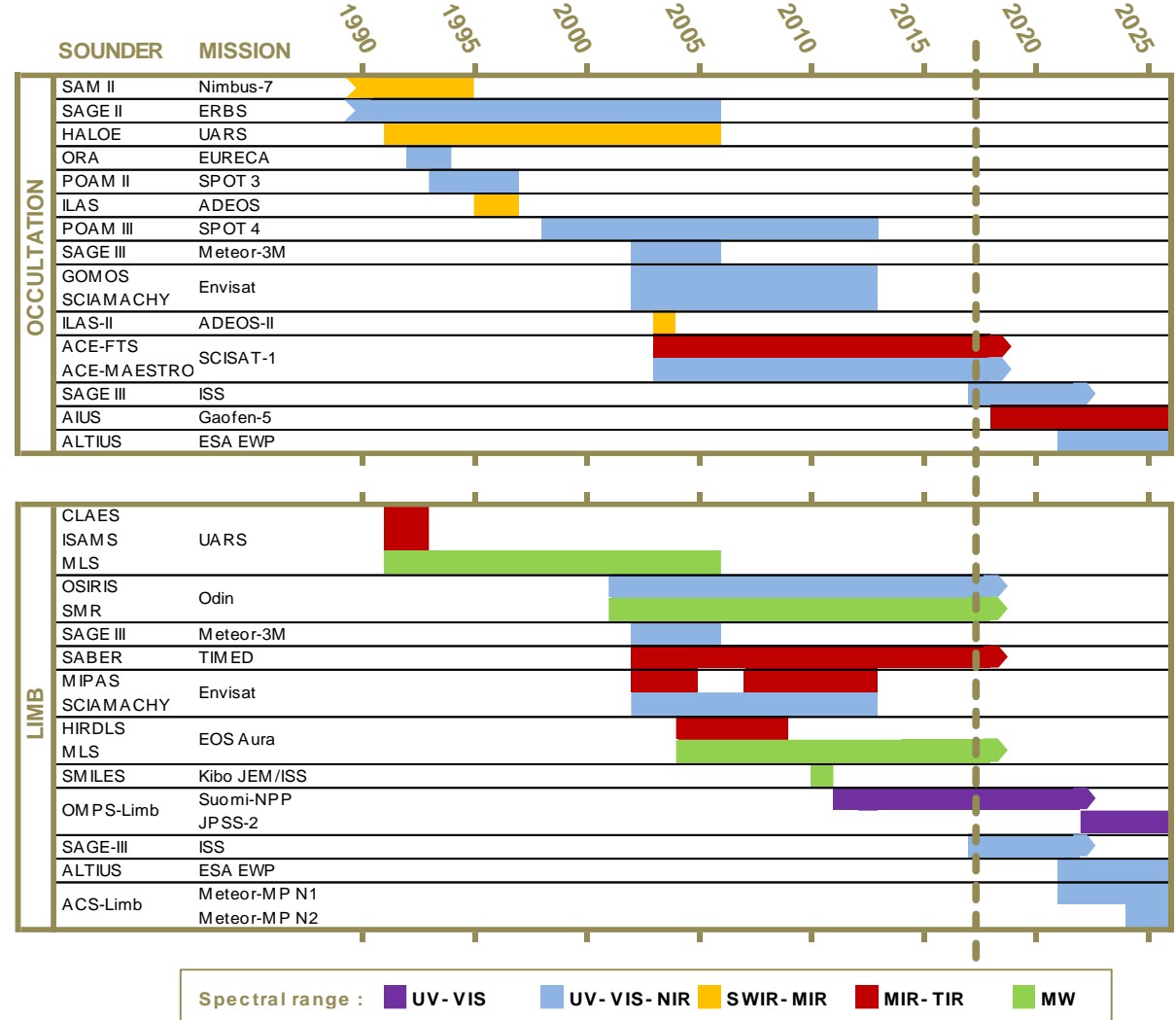

**Fig. 6b: Low Earth Orbit limb-viewing satellite profilers supported by NDACC, from the inception of the NDSC through present day and beyond. Upper chart: solar (and stellar) occultation instruments measuring in the UV / VIS / NIR (typically $O_3$, $NO_2$, BrO, $H_2O$, aerosols…) and the MIR / TIR (typically $O_3$, NO, $NO_2$, $HNO_3$, $CH_4$, CO, $H_2O$, CFC-11, CFC-12, aerosols…) Lower chart: limb-scanning instruments measuring scattered light solar radiation (typically $O_3$, $NO_2$, $H_2O$, aerosols…), MIR / TIR emission (typically $O_3$, NO, $NO_2$, $HNO_3$, $N_2O$, $CH_4$, CO, $H_2O$, HCl, $ClONO_2$, CFC-11, CFC-12, aerosols…) and MW emission (typically $O_3$, HCl, ClO, $N_2O$, $H_2O$, $HNO_3$…)**

Given the complete overlap of speciation of the Canadian Space Agency's ACE-FTS/SCISAT and the NDACC FTIR network, the latter provided validation for a suite of gases that were published in a series of papers ($O_3$: Dupuy et al., 2009; HCl, HF, $CCl_3F$, $CClF_2$: Mahieu et al., 2008; $NO_2$ and NO: Kerzenmacher et al., 2008; $N_2O$: Strong et al., 2008; $HNO_3$, $ClONO_2$, $N_2O_5$: Wolff et al., 2008; CO: Clerbaux et al., 2008; $CH_4$: De Mazière et al., 2008). Figure **7** shows two examples of these satellite validation efforts where the altitude resolution of the FTIR can be isolated to accommodate the satellite sensitivity range. The left panel from Clerbaux et al. (2008) compares CO partial columns from $6.5 - 8.5$ km to $20 - 25$ km depending on station

and sensitivity. The right panel concerns $HNO_3$ partial columns from several stations (Wolff et al., 2008) in the altitude range from $14.6 - 16.0$ to $29.0 - 31.0$ km.

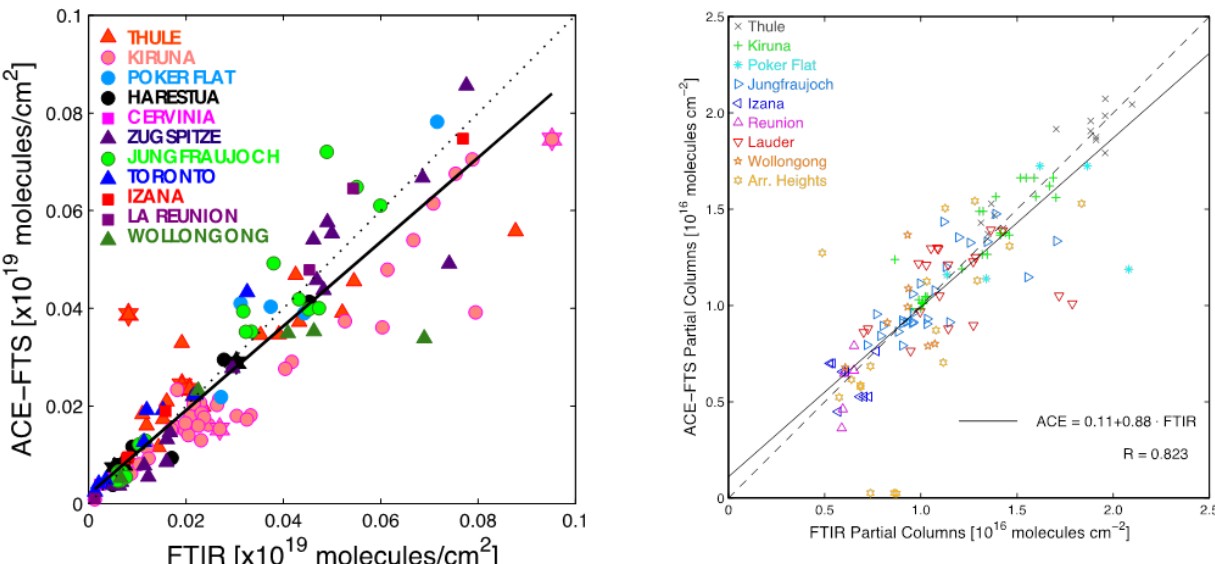

**Figure 7. Examples of the validation of new satellite datasets. The left panel is reprinted from [Clerbaux et al., 2008) for lower stratospheric CO measured at 5μm, from 11 instruments, 9 of which are located at NDACC stations. The partial column quantities from the ACE-FTS and the FTIR have very similar characteristics in vertical resolution and analysis providing a precise evaluation of satellite performance. The latitudinal extent of the NDACC sites here from 76ºN to 34ºS help verify the ACE-FTS global coverage. The right panel reprinted from [Wolff et al., 2008] shows the correlation of partial columns for HNO3 measured at 10.02μm from 9 NDACC stations from 76°N to 78ºS.**

NDACC lidar and sonde instruments provide insights into the upper troposphere/lower stratosphere (UT/LS) where several satellite measurements are less precise than in the middle and upper stratosphere. The capabilities of lidar and sondes are illustrated in Figs. 8 and 9. In Fig. 8(a) a day-long time-series of tropospheric ozone lidar variability is shown from the NASA/Goddard Space Flight Center (GSFC) Tropospheric Ozone Lidar (GSFC TROPOZ, Sullivan et al., 2014) alongside six ECC sondes at JPL-Table Mountain Facility [TMF] during the Southern California Ozone Observation Project (SCOOP). A comparison of the GSFC TROPOZ and JPL-TMF tropospheric lidar is presented in Fig. 8(b) for the final ECC sounding from Figure 8(a), indicating that both lidars are accurately representing the variability and gradients sampled during the sonde ascent in the lower free troposphere as well as in the UT/LS.

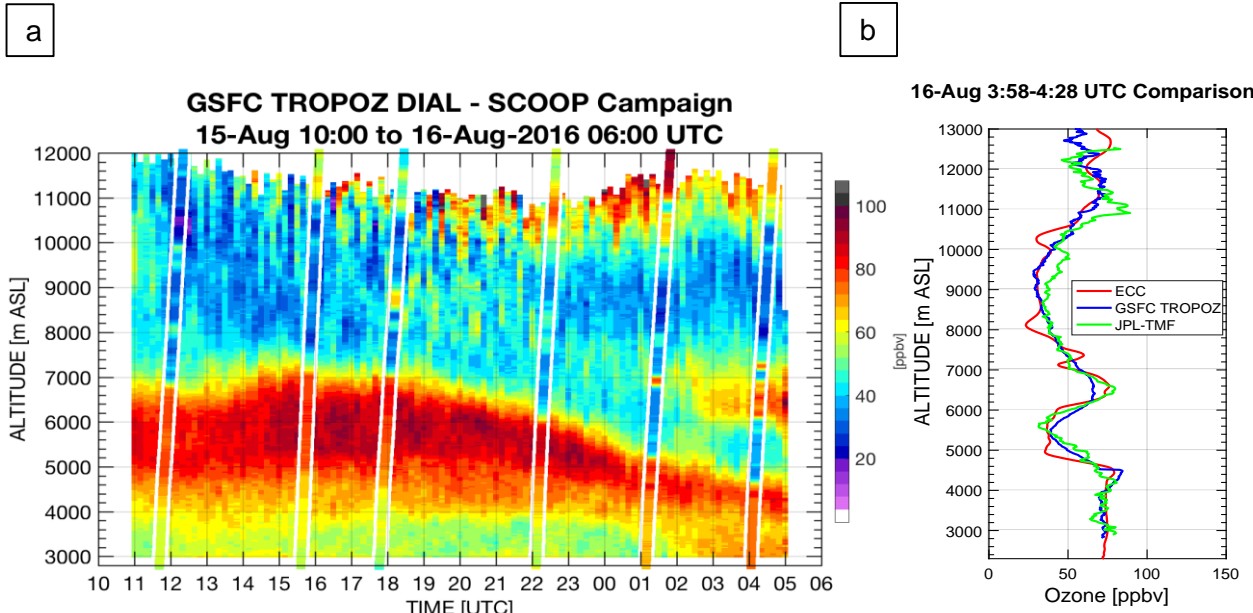

**Figure 8: a)** Time series of 10-min GSFC TROPOZ lidar observations during the SCOOP campaign at JPL-Table Mountain Facility (TMF, site elevation: 2300 masl) along with six ECC sondes (denoted with triangles). **b)** Comparison of 30-min averaged ozone from GSFC TROPOZ and JPL-TMF ozonelidars as compared to the last sounding of the time series. **Courtesy: J. T. Sullivan, NASA/GSFC.**

In Fig. 9, the longitudinal ozone structure in the TTL (tropical tropopause layer or tropopause transition layer, as the UT/LS is referred to in the tropics), is displayed using the composite tropical SHADOZ data (Thompson et al., 2003; 2012; 2017). The eastern Indian Ocean through Pacific region displays a sharp ozone gradient and high tropopause. The lower ozone values in the latter zone, relative to the South American-to-African region, in the center of Fig. 9, are attributed to more active convection in the western Pacific, where relatively unpolluted boundary-layer marine air is rapidly mixed upward. The fine structure of TTL ozone as observed with sondes serves as a reference for satellite retrievals and chemistry-climate models, in a region where ozone and temperature feedbacks are important.

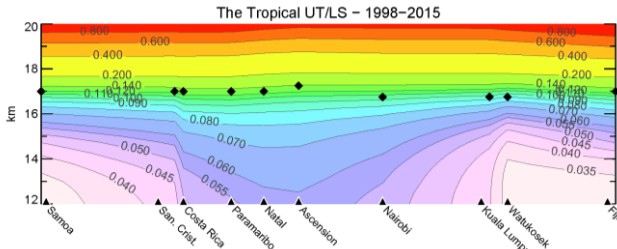

**Figure 9: TTL ozone structure (ozone contourlines in ppmv) from SHADOZ from 13-17.5 km above the stations, labeled by their host country, equatorward of 19°N/S based on all 1998-2015 data. Courtesy: J. C. Witte, NASA/GSFC.**

**3.4 Providing precise documentation of the multi-decadal trends of many tropospheric and stratospheric constituents.**

High-resolution solar absorption spectra regularly recorded by NDACC FTIR spectrometers under cloud-free conditions provide precise documentation of multi-decadal trends of many tropospheric and stratospheric constituents. For example, extended NDACC FTIR data sets, combined with HALOE observations from UARS gave evidence for a stabilization of stratospheric chlorine around the mid-1990s (Rinsland et al., 2003). Subsequently, NDACC showed there to be a decrease in atmospheric HCl and ClONO$_2$ at rates ~ 1%/year in both hemispheres, between 80°N and 78°S (Kohlhepp, et al., 2012). While it is believed this reversal is due to reduced emissions of anthropogenic source Cl species and that it will continue, note that the chlorine decline has not been monotonic since 1997 (Fig. 10). More recently, the NDACC FTIR time series provided evidence of a surprising re-increase in HCl in the Northern Hemisphere mid-latitude stratosphere of up to ~3%/year between 2007 and 2011. The cause of the HCl upturn was identified as being due to changes in atmospheric circulation (Mahieu et al., 2014). This is seen in Fig. 10, that shows the data sets (1983-2016) restricted to the June to November months; this limits the variability caused by atmospheric transport and subsidence mainly during winter-springtime. A good proxy of northern mid-latitude total inorganic chlorine (Cl$_y$) is obtained by summing the HCl and ClONO$_2$ total columns (blue triangles). The thin continuous lines correspond to non-parametric least square fits involving an integration time of about 3-years. Using the 1997.0 Cly column as reference and the bootstrap resampling tool of Gardiner et al. (2008), a mean post-peak rate of change of -(0.50±0.15) %/yr is obtained for the 1997-2016 time period.

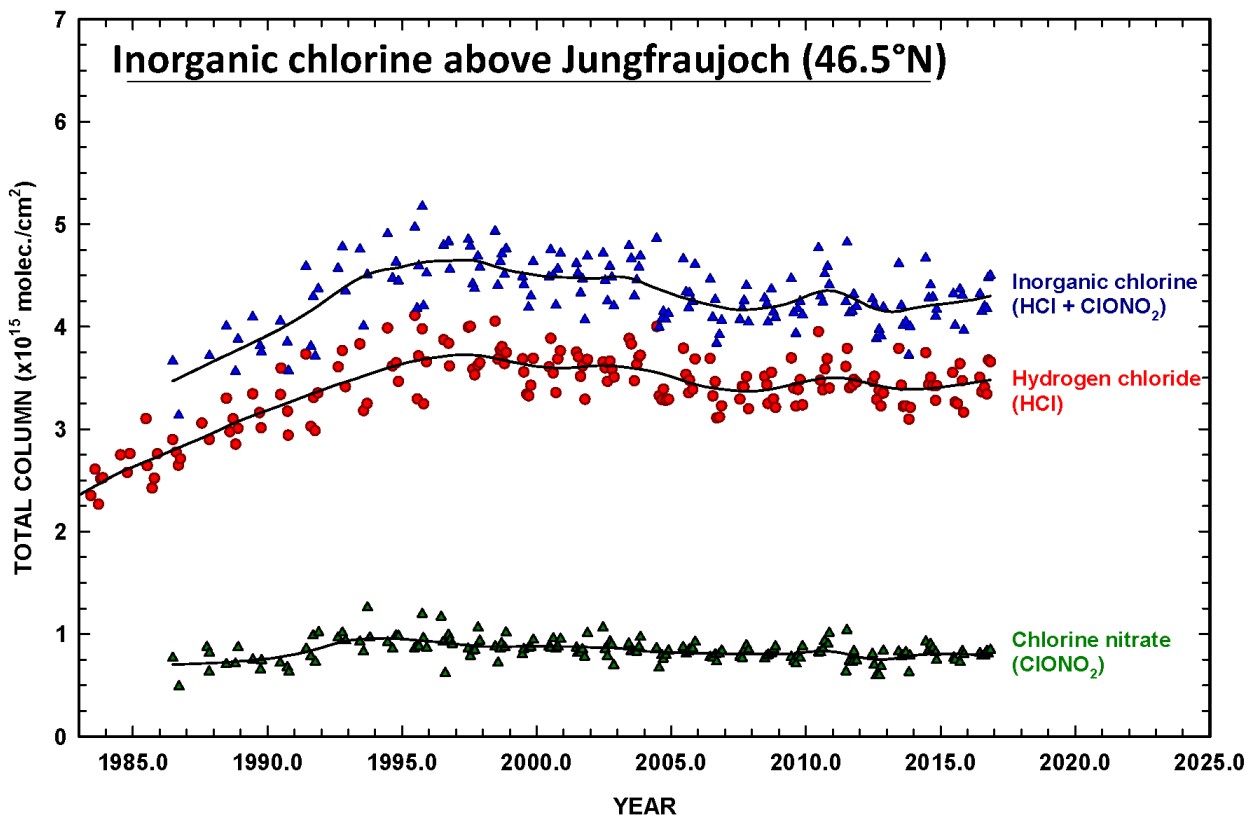

**Figure 10: Inorganic chlorine above Jungfraujoch. Multi-decadal monthly mean total column time series of the two main chlorine reservoirs, hydrogen chloride (HCl; red circles) and chlorine nitrate (ClONO2; green triangles), monitored at the Jungfraujoch station (Swiss Alps, 46.5°N, 3580 m a.s.l.) in the framework of the NDACC network. The data sets are restricted to the June to November months such as to limit the impact of variability caused by atmospheric transport and subsidence mainly during winter-springtime. A good proxy of northern midlatitude total inorganic chlorine (Cly) is obtained by summing the HCl and ClONO2 total columns (see blue triangles). The thin continuous lines correspond to non-parametric least square fits involving an integration time of about 3-year, they help appraising a non-monotonous decrease of chlorine in the stratosphere after 1996-1997. Using the 1997.0 Cly column as reference and the bootstrap resampling tool of Gardiner et al. (2008), a mean post-peak rate of change of -(0.50±0.15) %/yr is obtained for the 1997-2016 time period. Courtesy: E. Mahieu et al., Univ. Liège.**

NDACC microwave instruments have also provided evidence for decreasing stratospheric chlorine. Note that measurements of upper stratospheric ClO from Mauna Kea, Hawaii, showed a trend of -0.64±0.15%yr$^{-1}$ (2σ) from 1995 to 2012 (Connor et al., 2013), while microwave measurements of lower stratospheric ClO from Scott Base, Antarctica, during the ozone hole season suggest a trend in Cly of -0.6±0.4%yr$^{-1}$ from 1996-2015 (Nedoluha et al., 2016).

NDACC data are also noteworthy for filling gaps in satellite datasets. In Nedoluha et al. (2011), ground-based microwave measurements of upper stratospheric ClO were used to show that (within the specified errors) there was no reason to apply any bias correction in order to use UARS MLS measurements of ClO (1991-1998) and Aura MLS measurements of ClO (2004-present).

The primary halocarbon trace gases, both natural and anthropogenic, that are components of the ESC (Fig. 5) are largely measured through in-situ data collection at stations worldwide, mostly from the HATS and AGAGE cooperating networks

(Table 2). The effects of the Montreal Protocol and its follow-on amendments have been clearly observed. The finding that $CCl_4$ (carbon tetrachloride) has not declined in agreement with reported industry production data, led to the recent participation of the NDACC and SPARC communities in a targeted assessment of $CCl_4$ (www.wcrp-climate.org/WCRP Reports/2016/SPARC_Report7). Causes for the atmospheric budget disparity include under-reported industrial output,

fugitive sources and unintended manufacture due to numerous secondary reactions of Cl-containing compounds.

In Fig. 11 a newer application of NDACC FTIR data is shown. There is great interest in whether or not a surge in oil and natural gas (ONG) extraction by unconventional methods (tar sands, hydraulic fracturing or "fracking") is increasing burdens of non-methane hydrocarbons (NMHC or volatile organic compounds (VOC)) associated with ONG activity. Increases in

ethane column abundances over the period (2003-2015) as measured at five NDACC FTIR sites in Fig. 11 appear together with model interpretation (Franco et al., 2016). Rates of increase vary from ~+3%/yr at the remote Mauna Loa station but a little more than +5%/yr at mid-latitude continental locations, Toronto, Boulder and Jungfraujoch. Franco et al. (2016) have shown that an increase of the North American anthropogenic $C_2H_6$ emissions, dominated by up to 80% by emissions from the oil and gas sector, from 1.6 Tg yr−1 in 2008 to 2.8 Tg yr−1 in 2014, i.e. by 75%, is needed to capture the recent observed

rise in C2H6 atmospheric abundances.

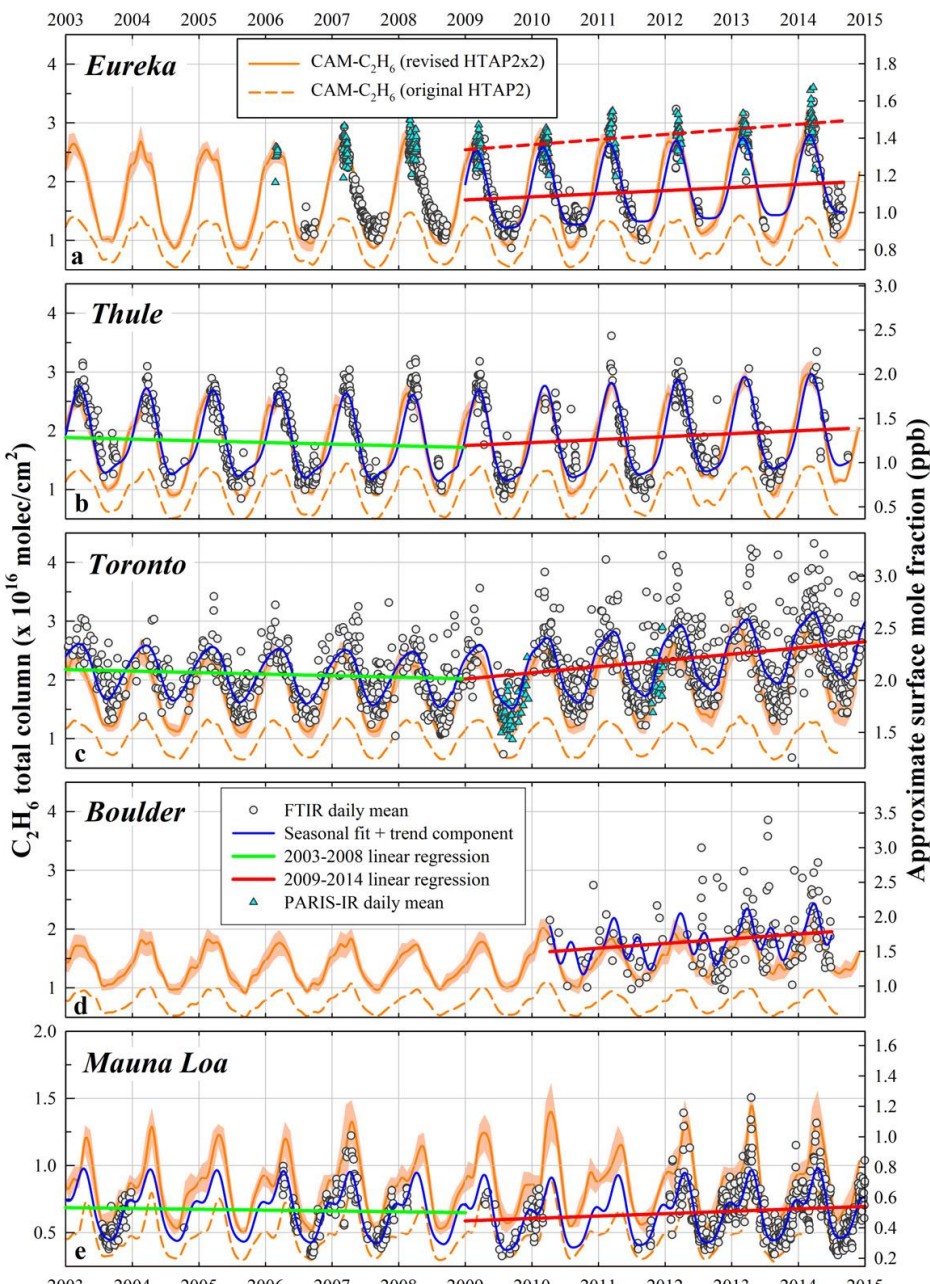

**Figure 11.** Daily mean $C_2H_6$ total columns derived from the NDACC FTIR (gray circles) and PARIS-IR (light blue triangles) observations between January 2003 and December 2014. The right y-axis scale converts the total columns into approximate surface mole fraction. The blue curve visualizes the function (including seasonal modulation and trend component) fitted to all daily FTIR means over the periods 2003–2008 and 2009–2014, using a bootstrap method. The green and red lines are the associated linear regressions (as solid line for FTIR and dashed line for PARIS-IR). The dashed and solid orange curves are the monthly mean C2H6 total columns simulated byCAM-C2H6, implementing the original HTAP2 and revised HTAP2 inventories, (scaled globally by a factor 2 and since 2008 by 20%/yr in N. America), respectively. The shaded area corresponds to the 1σ standard deviation.

**3.5 Understanding water vapor and assessing the measurement techniques.**

Figure 12 displays the change in water vapor over Mauna Loa, Hawaii, as measured by NDACC ground-based microwave measurements near the stratopause since 1996.  Nedoluha et al. (2013) showed that, since 2004, these interannual variations tracked closely with both the local variations measured by Aura MLS and those measured from 50°S-50°N, thus demonstrating the value of single-site measurements of water vapor in this region for understanding near global variations.  Together with long-term measurements of water vapor in the lower stratosphere from balloons (e.g., Hurst et al., 2011) the NDACC measurements are tracking the complex long-term changes in stratospheric water vapor. Having a reference for stratospheric water vapour changes from ground-based measurements and balloons will become particularly important in the future when there will be fewer, if any, available satellite measurements of water vapour. Meanwhile, the FTIR long-term data set on the variability in isotopic ratios of water (e.g., Barthlott et al., 2017) has become an important tool for investigating different water cycle processes that are important in Earth's climate system.

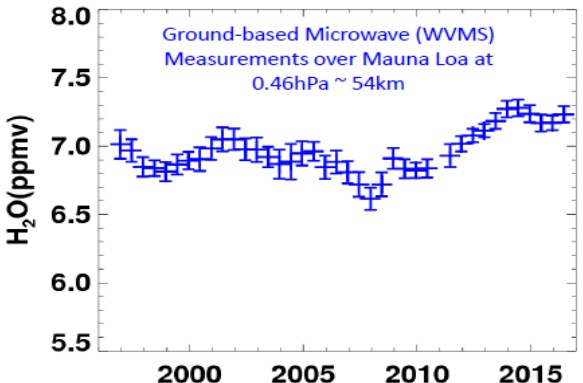

**Figure 12: Annual average water vapor mixing ratios at 54 km(~0.46 hPa)  over Mauna Loa (19.5°N, 204.4°E).  Symbols are shown for January–December and July–June; the seasonal cycle has been removed. Thus, each measurement is included in two annual anomalies.  Mixing ratios are retrieved from ~weekly integrated spectra; error bars represent the standard deviation of the mean relative to a seasonal climatology.**

**3.6 Latitudinal differences of UV-A and erythemal ("sunburning") radiation**

Latitudinal variations in the annual doses of UV-B (280 – 315 nm) and UV-A (315 – 400 nm) radiation have recently been assessed using data from NDACC UV spectroradiometers (Braathen, 2015; Bais et al., 2015). In Fig. 13, we present an expanded comparison of latitudinal differences between the annual UV-A dose and the annual erythemal dose, i.e., the UV dose on a horizontal surface causing sunburn (McKinlay and Diffey, 1987). Latitudinal gradients are stronger for the erythemal dose than the UV-A dose (Fig. 13(a)), partly because photons travel a longer path through the atmosphere for the lower solar elevations prevailing at higher latitudes, allowing greater absorption of UV-B radiation by ozone. The ratio of erythemal and

UV-A dose (Fig. 13(b)) is about a factor of two larger at the equator than near the poles. This latitudinal dependence is in accordance with earlier findings and similar to that of the ratio of UV-B / UV-A reported by Seckmeyer et al. (2008a) and Bais et al. (2015) because wavelengths in the UV-B range contribute about 90% to the erythemal dose.

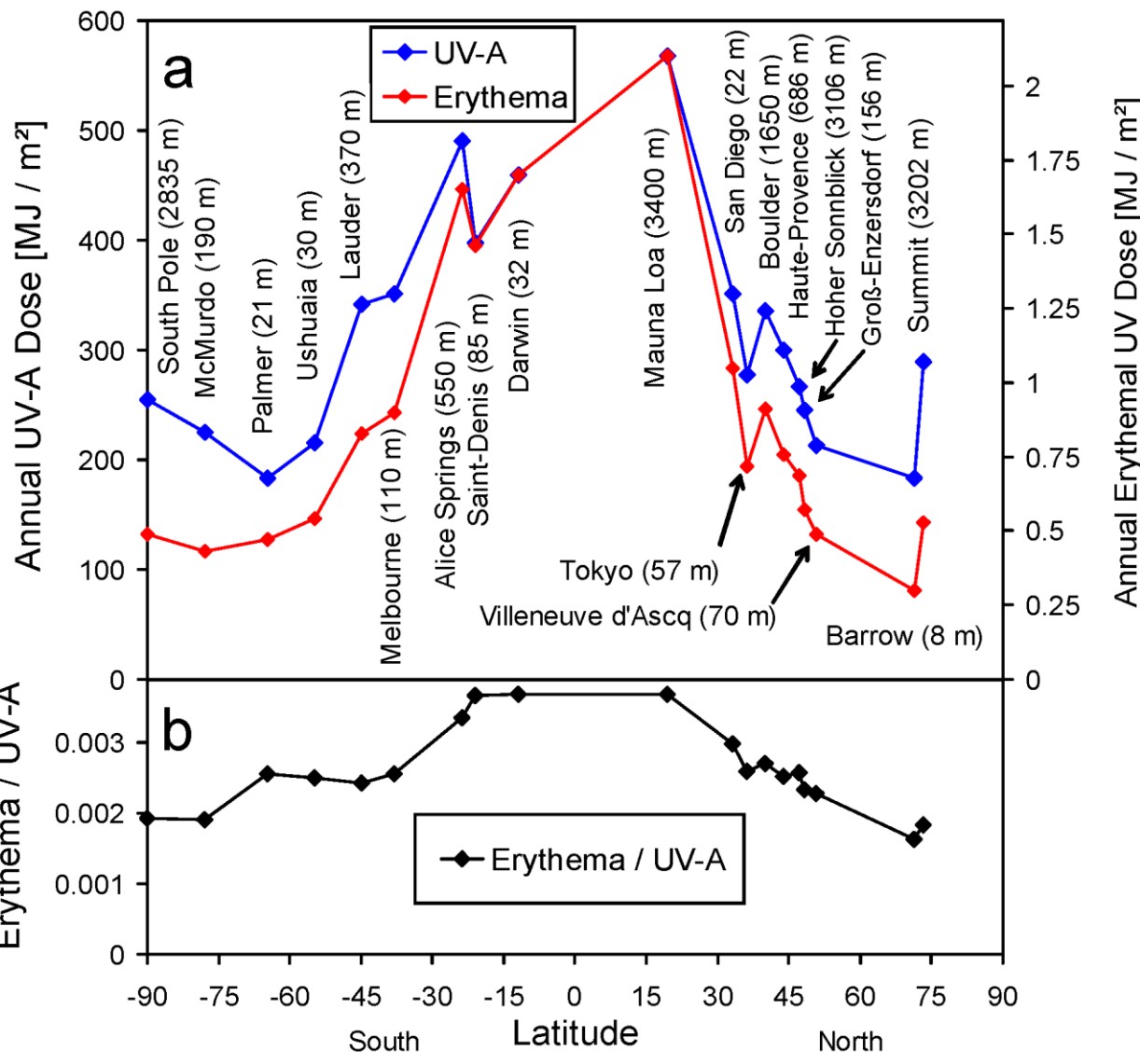

Figure 13: Latitudinal variation of UV-A and erythemal annual dose (a) on a horizontal surface, and (b) the ratio of erythema/UV-A dose. Note that high-altitude stations (South Pole, Mauna Loa, Boulder, Hoher Sonnblick, Summit) receive considerably higher erythemal and UV-A doses than stations closer to sea level (Groß-Enzersdorf, Barrow). Instruments at Melbourne, Darwin, San Diego, and Tokyo are not formally part of NDACC but use the same instrumentation and data processing methods as NDACC-affiliated stations.

Differences between corresponding latitudes in the Northern and Southern Hemispheres can be attributed to differences in cloudiness, total ozone, aerosol loading, Sun–Earth separation, altitude, and albedo (Seckmeyer et al., 2008b; Bais et al., 2015). The annual erythemal dose is approximately a factor of four larger in the tropics than at high latitudes. In the tropics, it reaches about 1.75 MJ m-2 near sea level. This corresponds to an average daily dose of 4,800 J m-2 (or 48 standard erythemal doses (SED)). For fair skinned individuals (skin type I), the minimal dose leading to reddening of the skin is about 200 J m-2 (Vanicek et al., 2000). Hence, the average daily dose at the equator is about 24 times the minimal erythemal dose (MED). Note that the maximum daily erythemal dose ever observed at Mauna Loa is 9,500 J m-2 (or 95 SED) (McKenzie, 2016).

In Antarctica, the prevailing low solar elevations are partly compensated by high surface albedo, 24 hours of sunlight in the summer, the effect of the ozone hole, and high surface elevation (Bernhard et al., 2010). Because of these factors, annual erythemal UV doses in Antarctica are still significant and within a factor of two of mid-latitude values. Fig 13(a) also indicates that high-altitude stations (South Pole, Mauna Loa, Boulder, Hoher Sonnblick, and Summit) receive considerably higher erythemal and UV-A doses than stations closer to sea level (for example, compare Hoher Sonnblick vs. Groß-Enzersdorf and Barrow vs. Summit). High surface reflectivity ranging from 0.96 to 1.00 also contributes to the relatively large doses at the South Pole and Summit (Bernhard et al., 2008), whereas attenuation of UV radiation by clouds and aerosols is responsible for the relatively low dose at Tokyo (McKenzie et al., 2008).

NDACC spectral UV measurements have also been used recently to validate surface UV levels derived from satellite observations, specifically from the Ozone Monitoring Instrument (OMI) on Aura and the Global Ozone Monitoring Experiment (GOME)-2 instrument on Metop-A (Brogniez et al., 2016).

### 3.7 Evaluating coupled chemistry-climate models

NDACC data have been used in the evaluation of coupled Chemistry–Climate Models (CCMs) under the CCMVal activity conducted by the SPARC project of the World Climate Research Program (WCRP) in which the radiative, dynamical, transport, and chemical processes in the models were analyzed in unprecedented detail. In particular, the long time series of NDACC column observations of HCl and ClONO$_2$ from Jungfraujoch (47oN) were used to evaluate simulated trends in stratospheric chlorine from 1990-2007. This comparison revealed unrealistically high and low chlorine levels in some models as well as differences in the simulated partitioning between these species. Douglass et al. [2014] also evaluated the CCMVal models using NDACC data and explained how problems with simulated chlorine impacted predictions of ozone recovery. By using column data from 7 NDACC stations spanning 68oN to 45oS, Douglass et al. determined that simulated total chlorine and the partitioning between HCl and ClONO$_2$ were controlled by lower stratospheric transport and thus models with the most realistic transport produced similar ozone projections. This study successfully explained the causal link between poor transport and the wide range of ozone recovery projections of the CCMs.

Model output generated by the Theory and Analysis Working Group are also now available at the website ftp://ftp.cpc.ncep.noaa.gov/ndacc/gmi_model_data . Model simulations that are integrated with reanalysis meteorology have realistic constituent variability from daily to seasonal timescales. Simulated station data are useful for providing an

understanding of station data variability and representativeness, thus building a bridge between individual station measurements and the global perspective. Model simulations produced by the group can be used to help set priorities for network expansion and/or instrument relocation.

Figure 14 shows how a simulation with the GMI chemistry transport model integrated with MERRA meteorological fields can

be used to understand sampling issues at a polar station (Kiruna, 68ºN, 20ºE). The top panel shows 255 FTIR $HNO_3$ measurements made during a 4-year period. Measurements are sparse in winter when $HNO_3$ columns are highest, leading to bias in calculated seasonal or annual trends. Simulated $HNO_3$ columns from GMI (black) are also shown for Kiruna and only on the same dates as the FTIR measurements. These show realistic seasonal and daily variability, demonstrating the simulation's value for estimating sampling biases. The bottom panel shows simulated $HNO_3$ columns for 'monthly' means

calculated only from the measurement dates (red), true monthly means (black), and the zonal monthly mean for the 65-70ºN latitude band (blue). Their differences indicate the Kiruna FTIR data most closely sample true monthly station means and true monthly zonal means in summer and fall, but in winter, sparse sampling and large dynamical variability in the Arctic lead to large negative biases, especially notable in early 2014. The mean difference is -5%.

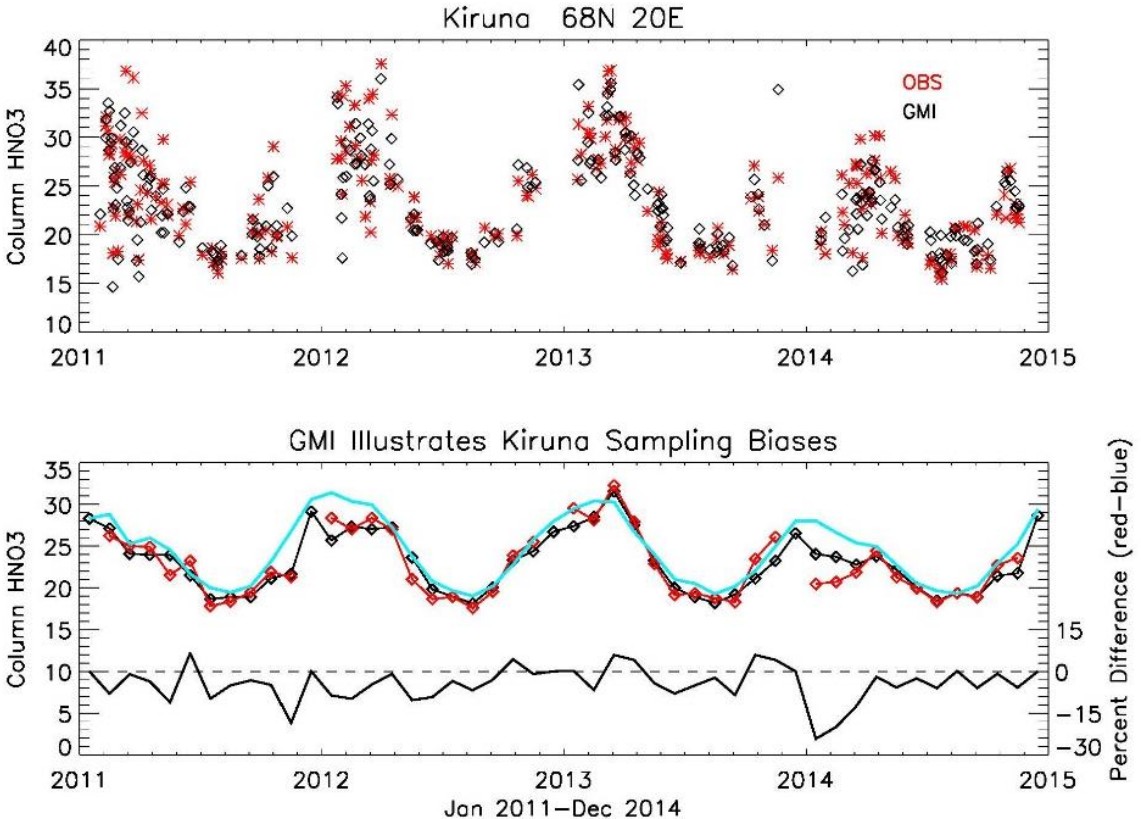

**Figure 14: Application of a chemical-transport model (GMI, Global Modeling Initiative) to interpolation of NDACC data from a polar station (Kiruna, 68ºN, 20ºE). The top panel shows 255 FTIR $HNO_3$ measurements. The lower panel shows how the model with MERRA analyses estimates the impact of sampling bias due to missing winter data.**

# 4 NDACC's position in the landscape of atmospheric monitoring networks

## 4.1 Complementarity among existing networks

As indicated above, NDACC recognizes, on the one hand, the complexity of the atmospheric system and the large variety of needs to appropriately monitor this system, and (2) the existence of a multitude of atmospheric monitoring networks, each of which have a particular focus and level of maturity. NDACC fills a particular niche in this landscape, with its focus on the long-term monitoring of the atmospheric composition (gases and particles) from the free troposphere to the lower mesosphere with dynamics (temperature and winds) for addressing the objectives outlined in Sect. 2.1. It uses essentially six ground-based remote-sensing techniques and sonde measurements, complemented by theoretical and modeling activities and satellite observations. NDACC further complements the cooperating long-term monitoring networks of in-situ atmospheric composition like AGAGE and HATS (see Table 2). By contributing stratospheric aerosol measurement, NDACC augments Earlinet (European Lidar Network) and MPLnet that have their focus on tropospheric aerosol.

NDACC and its Cooperating Networks make synergistic use of data from all the networks, thus benefitting from each other's expertise in addressing scientific questions, and identifying common issues, e.g., spectroscopic requirements, needs for infrastructures for digital services in terms of networking, computing and data management, reporting guidelines, etc.

## 4.2 Tiered system of systems

The various existing networks can be considered a tiered system of systems as outlined in Thorne et al. (2017). In this system, the networks are categorized as reference, baseline or comprehensive, depending on a number of measurement maturity criteria for the observations, the reported data and their availability and network protocols. The scoring for the different maturity criteria is represented in a maturity matrix.

The application of the maturity concept to the NDACC network as a whole is shown in Fig. 15. It must be emphasized that the scores should be taken with an uncertainty margin of plus/minus 1. The maturity matrix shows that NDACC satisfies the requirements of a global reference network at several sites, which means that it provides metrologically traceable observations, with quantified uncertainties, at a limited number of locations with quasi-global coverage.

Because of its maturity, NDACC is among the networks that are recognized by the European Copernicus initiative as key for providing data for validation of the Copernicus Atmosphere Monitoring Service (CAMS) (https://atmosphere.copernicus.eu/user-support/validation/verification-global-services) and for providing reference ground-based data in the Copernicus Climate Change Service ( C3S_311A_LOT3: Access to Observations from Baseline and Reference Networks). Similarly, it is among the key networks providing Fiducial Reference Measurements (FRM) for satellite systems. This network role includes providing independent data for the validation of satellite climate data records for ozone in the ESA Climate Change Initiative (CCI) (http://www.esa-ozone-cci.org/).

Work is continuously ongoing to improve the reference quality of NDACC data; this work is supported in part by ESA in its FRM programme, e.g., for the UV-visible DOAS-type measurements.

| Metadata | Documentation | Uncertainty characterization | Public access, feedback and update | Usage | Sustainability | Software (optional) |
|---|---|---|---|---|---|---|
| Standards | Formal Description of Measurement Methodology | Traceability | Access | Research | Siting environment | Coding standards |
| Collection level | Formal Validation Report | Comparability | User feedback mechanism | Public and commercial exploitation | Scientific and expert support | Software documentation |
| File level | Formal Measurement Series User Guidance | Uncertainty Quantification | Updates to record | | Programmatic support | Portability and numerical reproducability |
| | | Routine Quality Management | Version control | | | Security |
| | | | Long term data preservation | | | |

Legend

| 1 | 2 | 3 | 4 | 5 | 6 | Not applicable |
|---|---|---|---|---|---|---|

comprehensive      baseline

Figure 15: Maturity matrix for NDACC as a whole. The column headers indicate different assessment strands; the different cells in the columns indicate several assessment categories in the strand. The colour scale indicates the maturity score, according to the levels indicated in the bottom line.  Courtesy: P. Thorne et al. (2017) .

## 5 Recent evolution of NDACC and challenges

### 5.1 Measurement Strategies

#### 5.1.1 Quality Assurance

Since its operational start in 1991, NDACC has paid great attention to ensuring the quality of the individual data as well as the consistency of the data throughout the network. The expansion of the network, as well as the scientific questions that the community is addressing (e.g., the need for precise ozone trend estimates, the use of NDACC data for satellite validation, etc), have intensified the quality requirements. Several working groups have established strategies to better ensure station consistency in operations (e.g., Peters et al, 2017) to deal with uncertainty estimations (e.g., Leblanc et al., 2016a-c) and to better document data through reporting guidelines and traceability requirements. The transition from the NASA Ames format

to the GEOMS HDF format for data reporting and archiving (see Sect. 2.3) supports efforts for better documentation and traceability. Additional efforts are underway, with support from the European Union in the Copernicus framework, to improve the quality and consistency of the data reporting in the GEOMS HDF format, which will result in an enhanced accessibility and quality of the NDACC DHF. Also a tool to enable the user to convert the data from GEOMS HDF to NetCDF will be made available on the NDACC DHF.

Network quality control and site-to-site consistency have been achieved through different methods. Intercomparison campaigns were a primary method. These usually gather many instruments, e.g., UV-VIS spectrometers in the Cabauw campaigns in a single location (Piters et al., 2012). In the case of the Infrared working group (IRWG), side by side deployment of several instruments including a mobile FTIR instrument was made to compare instruments and to harmonize operating procedures (e.g. Goldman et al., 1999). Mobile reference instruments are used for cross-calibrating stratospheric lidar (Godin et al., 1999; Keckhut et al., 2004; Steinbrecht et al., 2009) and to maintain rigorous standards within the Dobson network (Komhyr et al., 1989; Evans et al., 2017). For ozonesondes the World Centre for Calibration of Ozonesondes in Jülich operates a chamber in which several instruments are tested simultaneously with ozone, temperature and pressure profiles that simulate environments from tropical to polar conditions (Smit et al., 2007). These intercomparisons typically include a blind analysis phase when one or more outside referees review the data ahead of the instrument investigator team. At times intercomparisons become inefficient or impractical. In that case, for spectrometric methods, quality checks can be based more heavily on cell measurements (Hase, 2012) and continued retrieval intercomparisons. Quality checks using XCO2 retrievals have been suggested by Barthlott et al. (2012) where a correlation among several FTIR sites shows a very good site-by-site consistency. In addition, NDACC has offered a framework for the evaluation of retrieval algorithms, e.g. the work by Leblanc et al. (1998) in examining and comparing lidar temperature retrieval algorithms using simulated data, and subsequent work standardizing resolution and error budgets in temperature, ozone and water vapor measurements with lidar (Leblanc et al., 2016a-c).

Several NDACC instrument working groups are considering centralized data processing in order to avoid inconsistencies among the individual stations/partners that originate in differences in data processing software. In some instrument working groups, standard data processing software is already used by all partners (give example), but this does not completely avoid discrepancies due to software being implemented in a different way or used with different parameters/settings. Transitioning to standard data processing software or centralized data processing is challenging because NDACC has been and remains a research-oriented network, in which some instruments have been uniquely developed by the Principal Investigator (PI) with customized data processing codes. This is especially the case when the PIs play essential roles in the reporting of the data and have worked to assure their data quality, uncertainties and network consistency through data intercomparison campaigns (e.g., Deshler et al., 2008; Piters et al., 2012; 2017; Sterling et al., 2017; Thompson et al., 2017; Witte et al., 2017). Nevertheless, in cases where there is a need for a more operational data delivery, e.g., for satellite and model validation purposes, efforts are underway to set up prototype centralized processing systems, e.g., in the UV-visible Working Group with support from ESA. These quality assurance efforts can be important in elevating the maturity level of NDACC network, as shown in Fig. 15.

### 5.1.2 Rapid Data Delivery.

NDACC Instrument PIs have always been encouraged to deliver data as soon as possible. Recently, in the spirit of more operational data delivery, the timeframe for offering NDACC data to the public has been shortened from two years to one year after acquisition. Additionally, a separate section has been created in the data archive, called the Rapid Delivery (RD) database (ftp://ftp.cpc.ncep.noaa.gov/ndacc/RD/) where data users can find preliminary NDACC data and data from candidate stations; these data have not yet been completely quality-controlled but are sufficiently reliable for supporting at least preliminary satellite or model validation.

## 5.2 Challenges

### 5.2.1 Continuity of measurements

The maintenance of trend-quality stable measurements over the past 25 years, often under inhospitable environmental conditions, and the operation of such instruments over future decadal timescales, is, and will continue to be, a daunting challenge. Stable measurements require that aging instrumental components be replaced on a regular basis, and, as technologies become obsolete, upgraded components must be deployed. Upgrading instruments can both improve measurement quality and allow for continued operation under tight fiscal constraints. Making such transitions while providing the community with stable data sets will continue to require careful engineering, intercomparisons and measurement evaluation.

In addition to the scientific and engineering challenges of long-term measurements, there are fiscal challenges of maintaining such support in an ever changing budget environment. Even a brief gap in funding imperils the continuity that is crucial for determining long-term trends, which is the fundamental goal of NDACC. Changing scientific priorities may shift away from long-term ground-based measurement programmes, often towards space-borne platforms. In the latter case, space agencies may not recognize their strong dependence on NDACC-type data for validation and assume that other scientific sponsors will provide the necessary long-term financial support.

### 5.2.2 Role in scientific assessments

The standard complement of NDACC instruments/sites should be considered as an essential part of a research infrastructure that delivers high-quality data for atmospheric parameters, trace gases and aerosols to the scientific community and to the policy makers, for a multitude of purposes. These infrastructures, including instrument maintenance (upgrading, cross-calibration, etc.) and the data they deliver, deserve continuous support from the stakeholders to ensure the fulfillment of the research and to provide the essential scientific basis for environmental policies. For example, the current threats associated with climate change require continuous, long-term high-quality monitoring and reporting of the state of the atmosphere, including its chemical composition, analogous to the obligations that many nations have assumed for air quality monitoring and reporting.

Indeed, an important lesson learned from NDACC is the necessity of having multiple and independent long-term records. The high level of accuracy and stability needed to observe small and slow changes in the atmosphere rests on comparing a number of different instruments and techniques. Only with such data can the community support the Intergovernmental Panel on Climate Change in assessing the current state of the Earth's climate and for ensuring that mitigation and adaptation

options are rooted in high-quality observations. The same holds true for the WMO/UNEP Scientific Assessments of Ozone Depletion: e.g., the lack of plans to fly limb satellite sounders operationally highlights the urgent need to maintain a strong ground-based measurement system so to be able to support the WMO in its task to assess the state of the ozone layer every four years as requested by the Montreal Protocol.

## 6 Concluding Perspectives

NDACC is transitioning to a network that is both research-oriented and operationally-oriented, providing data and analyses to a large variety of users: researchers, large-scale initiatives like Copernicus, space agencies of many countries, policy-oriented assessments, and the public at large. These data users rely on NDACC remaining healthy, with a well-supported infrastructure and with a dedicated operational infrastructure (i.e., the community of scientific experts) that updates measurement capabilities to meet new data needs.

However, this evolution must not hinder further development of the network for pure research purposes, which in the longer term will also serve the other users.

Some of the future developments envisaged in NDACC include:

(1) filling important gaps in the network spatial and temporal coverage, i.e., there are currently few stations in the tropics, notably in South America, Asia, Africa. Few observations cover the full diurnal cycle; this will be essential for the

20           validation of geostationary satellites. Model-based network design will help to identify where such coverage (spatial and temporal) gaps lie.

(2) filling important gaps in the ensemble of atmospheric variables that are observed (e.g., as new ODS-substitute products are released by human activities, it is important to monitor their fate and their evolution in the atmosphere).

(3) refining existing and/or developing new measurement techniques to improve the accuracy, precision and traceability

25           of the data products.

(4) automating operations (observations, data processing, quality assurance/quality control (QA/QC), etc.) where possible to lower costs.

(5) developing more compact, mobile, and less-expensive instruments, to enhance their deployment in developing countries, remote locations and for campaign purposes.

(6) closer work with the modeling community to evaluate chemistry-climate modules (cf. the Chemistry-Climate Model Initiative (CCMI) (Morgenstern et al., 2017)) and chemical-transport models.

(7) providing early warning of volcanic eruptions using various NDACC instruments, especially in conditions of compromised satellite observations.

In all of these activities, NDACC is committed to interaction with a range of user communities who recognize the value of ground-based observations.

*Acknowledgments* Since its early years, NDSC/NDACC has received outstanding administrative support from Kathy A. Thompson (CSC, SSAI). US support for NDSC/NDACC instruments and sites, construction and maintenance of the database, and for some of the Cooperating Networks, has been provided by NASA through UARP and related programs (Michael J. Kurylo and Kenneth W. Jucks, Program Managers), and by NOAA/OGP, NOAA/ESRL and NOAA/NCDC (now NCEI). In Europe, NDSC/NDACC activities have been supported through the European Commission Framework Programs. The NDSC/NDACC PIs are also grateful to ESA for support for dedicated campaigns and satellite validation, and to their national funding authorities. Some support has also been received from JAXA and Japanese authorities for NDACC sites in Japan and S. America (Rio Gallegos), and for validation of Japanese satellite missions. While the co-authors provided significant inputs to this paper, they thank all NDACC PIs and all current and past members of the NDACC Steering Committee listed herewith for their important efforts dedicated to the Network: V. Savastiouk, T. Blumenstock, N. Kämpfer, M. Shiotani, D. F. Hurst, B. Johnson, R. Stübi, B.-M. Sinnhuber, K. Kreher, M. Van Roozendael, R.G. Prinn, H. Vömel, H. Maring, R.F. Weiss, G. König-Langlo, C. Long, G. E. Bodeker, R. Dirksen, P.W. Thorne, J. Elkins, J. Notholt, P.O. Wennberg, P.V. Johnston, J.-P. Pommereau, B. Bojkov, A. Dehn, J.-F. Doussin, J.R. Drummond, S. Godin-Beekmann, A.N. Gruzdev, F. Immler, N. Larsen, E. Mahieu, A. Mizuno, H. Schmithüsen, R.C. Schnell, P. von der Gathen, R. Sussmann, O. Schrems, F. Cairo, A.J. Miller, R. Zander, I. S. McDermid. They also thank Colette Brogniez (Université Lille 1 − Sciences et Technologies, Villeneuve d'Ascq, France), Richard L. McKenzie (National Institute of Water & Atmospheric Research, NIWA Lauder, New Zealand), and Stana Simic (University of Natural Resources and Life Sciences, Vienna, Austria) who have contributed to Section 3.6 and provided data for Figure 13.

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

# Annex A

| Authors | Title | Working Group | Domain | Theme | Topic |
|---|---|---|---|---|---|
| Bader et al. | The recent increase of atmospheric methane from 10 years of ground-based NDACC FTIR observations since 2005 | IR | troposphere & stratosphere | CH4 | trends |
| Barthlott et al. | Tropospheric water vapor isotopologue data (H216O, H218O, and HD16O) as obtained from NDACC/FTIR solar absorption spectra | IR | troposphere | water vapor | technique |
| Baylon et al. | Background CO2 levels and error analysis from ground-based solar absorption IR measurements in central Mexico | IR | troposphere | CO2 | technique |
| Blanchard et al. | Thin ice clouds in the Arctic: cloud optical depth and particle size retrieved from ground-based thermal infrared radiometry | IR | troposphere | clouds | technique |
| Brogniez et al. | Validation of satellite-based noontime UVI with NDACC ground-based instruments: influence of topography, environment and satellite overpass time | spectral UV | troposphere & stratosphere | UVI | validation |
| Buchholz et al. | Validation of MOPITT carbon monoxide using ground-based Fourier transform infrared spectrometer data from NDACC | IR | troposphere & stratosphere | CO | validation |
| Christiansen et al. | Trends and annual cycles in soundings of Arctic tropospheric ozone | sonde | troposphere | ozone | trends |
| Deshler et al. | Methods to homogenize electrochemical concentration cell (ECC) ozonesonde measurements across changes in sensing solution concentration or ozonesonde manufacturer | sonde | troposphere & stratosphere | ozone | technique |
| Douglass et al. | Multi-decadal records of stratospheric composition and their relationship to stratospheric circulation change | Theory & Analysis | stratosphere | circulation | trends |
| Duflot et al. | Tropospheric ozone profiles by DIAL at Maïdo Observatory (Reunion Island): system description, instrumental performance and result comparison with ozone external data set | Lidar | troposphere | ozone | technique |
| Evans et al. | Technical note: The US Dobson station network data record prior to 2015, re-evaluation of NDACC and WOUDC archived records with WinDobson processing software | Brewer/Dobson | troposphere & stratosphere | ozone | technique |
| Fernandez et al. | Results from the validation campaign of the ozone radiometer GROMOS-C at the NDACC station of Réunion island | microwave | stratosphere | ozone | validation |
| Granados-Munoz et al. | Tropospheric ozone seasonal and long-term variability as seen by lidar and surface measurements at the JPL-Table Mountain Facility, California | lidar | troposphere | ozone | trends |
| Hausmann et al. | A decadal time series of water vapor and D / H isotope ratios above Zugspitze: transport patterns to central Europe | IR | troposphere | water vapor | trends |
| Hausmann et al. | Contribution of oil and natural gas production to renewed increase in atmospheric methane (2007–2014): top–down estimate from ethane and methane column observations | IR | troposphere | methane | trends |
| Khaykin et al. | Variability and evolution of the midlatitude stratospheric aerosol budget from 22 years of ground-based lidar and satellite observations | lidar | stratosphere | aerosol | trends |
| Kiel et al. | Comparison of XCO abundances from the Total Carbon Column Observing Network and the Network for the Detection of Atmospheric Composition Change measured in Karlsruhe | IR | troposphere & stratosphere | CO | validation |
| Knepp et al. | Intercomparison of Pandora Stratospheric NO2 Slant Column Product with the NIWA M07 NDACC Standard | UVVIS | stratosphere | NO2 | validation |
| Leblanc et al. | Proposed standardized definitions for vertical resolution and uncertainty in the NDACC lidar ozone and temperature algorithms – Part 3: Temperature uncertainty budget | lidar | stratosphere | temperature | technique |

| Leblanc et al. | Proposed standardized definitions for vertical resolution and uncertainty in the NDACC lidar ozone and temperature algorithms – Part 2: Ozone DIAL uncertainty budget | lidar | stratosphere | ozone | technique |
|---|---|---|---|---|---|
| Leblanc et al. | Proposed standardized definitions for vertical resolution and uncertainty in the NDACC lidar ozone and temperature algorithms – Part 1: Vertical resolution | lidar | stratosphere | ozone & temperature | technique |
| Mevi et al. | VESPA-22: a ground-based microwave spectrometer for long-term measurements of Polar stratospheric water vapor | Micxrowave | stratosphere | water vapor | technique |
| Moreira et al. | Comparison of ozone profiles and influences from the tertiary ozone maximum in the night-to-day ratio above Switzerland | Micxrowave | stratosphere | ozone | variability |
| Moreira et al. | The natural oscillations in stratospheric ozone observed by the GROMOS microwave radiometer at the NDACC station Bern | microwave | stratosphere | ozone | variability |
| Nedoluha et al. | 20 years of ClO measurements in the Antarctic lower stratosphere | microwave | stratosphere | ClO | trends |
| Peters et al. | Investigating differences in DOAS retrieval codes using MAD-CAT campaign data | UVVIS | troposphere & stratosphere | NO2 | technique |
| Reichert et al. | The Zugspitze radiative closure experiment for quantifying water vapor absorption over the terrestrial and solar infrared – Part 2: Accurate calibration of high spectral-resolution infrared measurements of surface solar radiation | IR | troposphere | water vapor | validation |
| Reichert et al. | The Zugspitze radiative closure experiment for quantifying water vapor absorption over the terrestrial and solar infrared – Part 3: Quantification of the mid- and near-infrared water vapor continuum in the 2500 to 7800 cm−1 spectral range under atmospheric condition | IR | troposphere | water vapor | validation |
| Steinbrecht et al. | An update on ozone profile trends for the period 2000 to 2016 | NDACC | stratosphere | ozone | trends |
| Sussmann et al. | The Zugspitze radiative closure experiment for quantifying water vapor absorption over the terrestrial and solar infrared – Part 1: Setup, uncertainty analysis, and assessment of far-infrared water vapor continuum | IR | troposphere | water vapor | validation |
| Toon et al. | Measurements of atmospheric ethene by solar absorption FTIR spectrometry | IR | troposphere | ethene | trends |
| Toon et al. | Atmospheric Carbonyl Sulphide (OCS) measured remotely by FTIR solar absorption spectrometr | IR | troposphere & stratosphere | OCS | trends |
| Van Malderen et al. | On instrumental errors and related correction strategies of ozonesondes: possible effect on calculated ozone trends for the nearby sites Uccle and De Bilt | sonde | troposphere & stratosphere | ozone | technique |
| Virolainen et al. | Quality assessment of integrated water vapor measurements at St. Petersburg site, Russia: FTIR vs. MW and GPS techniques | NDACC | troposphere & stratosphere | water vapor | validation |
| Weaver et al. | Intercomparison of atmospheric water vapor measurements at a Canadian High Arctic site | NDACC | troposphere & stratosphere | water vapor | validation |
| Yela et al. | Hemispheric asymmetry in stratospheric NO2 trends | UVVIS | stratosphere | NO2 | variability |
| Zeng et al. | Attribution of recent ozone changes in the Southern Hemisphere mid-latitudes using statistical analysis and chemistry–climate model simulations | Theory & Analysis | troposphere & stratosphere | ozone | trends |
| Zhao et al. | Accuracy, precision, and temperature dependence of Pandora total ozone measurements estimated from a comparison with the Brewer triad in Toronto | UVVIS | troposphere & stratosphere | ozone | validation |
| Zhou et al. | CFC-11, CFC-12 and HCFC-22 ground-based remote sensing FTIR measurements at Réunion Island and comparisons with MIPAS/ENVISAT data | IR | troposphere | (H)CFC | validation |