# Peer review of "The Network for the Detection of Atmospheric Composition Change (NDACC): History, status and perspectives"

_Atmospheric Chemistry and Physics, 2017_

## Referee Comment (RC1) · Anonymous Referee #1 · 18 Sep 2017

This manuscript presents a very valuable overview of the NDACC measurement Network, its history, structure, mission, and achievements. It is mostly well written, however in parts the content could be somewhat more logically organized and sometimes more factual. I found the information provided sometimes also confusing to what extent scientific results were really attributable to NDACC measurements and where not (e.g., Figure 11). I hence recommend publication after minor (although extended) revisions as detailed below.

Two more general criticisms that I have is that the authors should provide more detailed information on the kind of measurements that belong to NDACC and how they are

[Figure]

distinct from e.g. the WOUDC data archive, AGAGE, HATS, or other globally organised networks from NOAA. This information would be important to provide already in the abstract and introduction to put the network's measurements into better context with what else is going on. A better integration of all the NDACC-related activities in the diagrams and figures presented would thereby also be helpful. Secondly, since the papers main purpose is arguably not to provide new science but to be informative (namely on the background of NDACC) the paper would be more educational (and valuable) to the reader if it better referred to science papers that underpin the historical and scientific arguments presented.

Minor comments

Abstract The mission of NDACC does not become clear to me when reading the abstract. Is it mainly based on instruments and measurements that measure primarily stratospheric composition, or is it also focused on tropospheric composition? It may be good to list the different instruments (FTIR, lidar, ozonesondes, DOAS, Brewers, Dobsons. . .) that are actually part of NDACC (maybe in L28) so that the reader (if interested in a particular one of these) is informed right away.

Introduction Although the history of air quality research is being mentioned in the introduction, it remains unclear if any NDACC related measurements are contributing to studying this issue. I would state somewhere more explicit whether NDACC measurements concentrate primarily on the composition of the middle atmosphere (or not if a contribution to air quality observations is made).

P2L10 '. . .like carbon dioxide, methane. . .' do you refer to ground-based in-situ measurements here?

P2L16 A reference to Molina and Rowland, 1974 is here also needed.

Molina, M. J., and F. S. Rowland, Stratospheric sink for chlorofluoromethanes: Chlorine atom catalyzed destruction of ozone, Nature, 249, 820–812, 1974.

P2L17/19/22/23 More references are needed after all these statements. This overview paper otherwise misses a chance to improve its educational value.

P2L29 Please provide reference to this report or where it can be found.

P3L24 Add reference.

P3L27 Add 'US' after 'Colorado'

P4 L9 Please introduce when the name change came along, the mentioning of NDACC for the first time in the introduction is otherwise too abrupt.

P7L2 Wording is unclear. Are there other measurements not included in this figure that also belong to NDACC, or are all measurements included shown?

Figure 2 Do you show here affiliated stations only? It would be good to have the candidate stations included as well. I ask because I wonder whether there is really no candidate station in India? I also thought that there I one or even two NDACC stations in Africa (Kenia and South Africa)? If true, an update of this figure would be needed.

P9L2-4 Please rephrase sentence, I don't quite understand what you are driving at here. . .

P9L10-17 This mission statement of NDACC would seem more logical and appropriate at the very beginning of Section 2, or even already in the introduction.

Figure 3 Caption, why is the figure only similar, not equivalent to that in Figure 2? Also, I assume that theme and working groups are likely to overlap and it would be nice to reflect this in this diagram.

P11L1 This information sounds incomplete. Why do you only show three of the existing theme groups? It would be more valuable to provide the most up-to-date information even though this may mean that figure 3 would have to be updated.

P13L1 and elsewhere It is confusing to sometimes use NDACC only (even in title!) but

then in other places NDSC/NDACC. I would suggest to either always use one or the other throughout the paper, after where NDACC replaced the NDSC name.

P13L11 GOZCARDS should be listed in the list with SAGE/OSIRIS, SWOOSH, and ESA-CCI, since it is composed of the same kind of satellite instruments, not just after SBUV-MOD.

P13L13 It seems inappropriate to highlight the ground-based data here as being 'high-quality', since it implies the satellite measurements would not be high-quality. Please remove.

P13L13 It is confusing to include ozonesondes first, but then right away exclude them again, since they are not in the figure. Simply describe what is in the figure!

P13L17 Title would be more meaningful with 'Monitoring long-term atmospheric composition change'

P13L18-24 To my knowledge is it outdated to talk about the three stages of ozone recovery. If you want to stick to this, refer to the respective WMO ozone assessment in which these stages were discussed. Also, from more elaborate statistical evaluations than that presented here, it is not clear whether the second stage of ozone recovery is reached already. Please remove this statement or provide references as a proof.

Figure 5 caption 'ESC' is 'effective stratospheric chlorine', not what the authors indicate here. You may also want to add a reference to where this figure first appeared in a publication, I assume updated from WMO 2014?

P15L12/Figure 6 It's incorrect to say that figure 6 shows the limb satellite sounders. Please rewrite or produce a new figure.

P17L10 "NDACC excels..." is this not a little bit an overstatement? It may be true for any trace gas the lidars or ozonesondes may measure, but certainly for minor trace gas species, the resolution of the ACE-FTS satellite limb sounder (approx. 1-3 km in the UTLS) is not achieved?

P18L6 Please improve the English of this sentence.

Section heading 3.3 Suggest shortening to 'Constraining uncertainties in ozone absorption cross-sections'

Section 3.4 This section would be more appropriate/logical to move to right after (or included within) section 3.1. See suggestion of header change above.

Figure 11 Are AGAGE and HATS measurements really part of the NDACC system? If not, I would not tend to include this figure and discussion in the achievements section here. If yes, since you label it here EESC (in contrast to ESC in an earlier figure) it would be appropriate to explain to the general reader how these variables differ from each other.

Figure caption 12 I do not understand the information given in this figure caption and the main result that this figure represents. Please rework and be more specific also in the text.

P23L8 To have a reference on stratospheric water vapour changes from continuous ground-based measurements will become particularly important in the future and for future climate considerations, since past trends derived form merged limb satellite datasets have been yielding controversial results (cf. Hegglin et al., Nature Geoscience 2014).

P23L20 This definition sounds unnecessarily complicated to me, please consider rewriting.

Section 3.6 The title of this section is more complicated than necessary. Why do you say 'bounding factors', are these not direct UV measurements?

P25L23 What constituents have been evaluated in this context within CCMVal and in which papers? Please expand and provide references.

Section 4.2 I have to admit that I was somewhat lost reading the description of this

tiered system of systems concept by Thorne et al (2017). I assume a score of 1 is the worst mark, a score of 6 the best mark achievable? What are the requirements to achieve the label of a global reference network?

P28L14 What was the reasoning for using HDF? Wouldn't it have been easier for the community to use NetCDF as common format as mostly used in the climate modeling and Obs4MIP as well?

P30L16 suggest to add 'for validation', or did you refer to something else here?

P31L1 In fact, the lack of plans to fly limb satellite sounders operationally highlights the urgent need to maintain a strong ground-based measurement system so to be able to support the WMO in its task to assess the state of the ozone layer every 4 years as requested by the Montreal Protocol.

P31L25 Add reference to Morgenstern et al. (ACP, 2017)

Technical corrections

P4L24 Check punctuation.

P8L1 put comma after 'France'

P13L19 Missing bracket

Figure 9 caption change 'over the stations' to 'above the stations'

Figure 10 caption correct 'chorine' to 'chlorine'

Figure 12 caption correct 'Norther' to 'Northern'

Figure 14 caption remove extra space after 'Barrow'

---

## Referee Comment (RC2) · Anonymous Referee #2 · 19 Sep 2017

This paper by De Mazière et al., presents an overview of the long-term ground-based network for measurements of atmospheric composition on a global scale named NDSC (Network for the Detection of Stratospheric Change) at the beginning in 1991 then NDACC (Network for the Detection of Atmospheric Composition Change) since 2005. NDACC is focused on the chemical and physical state of the stratosphere and upper troposphere at global scale thanks to more than 80 stations distributed over the globe.

This paper is clear and well written. The introduction providing a brief history recalling the need for such international network is particularly pleasant to read. The following subsections aim to explain the organizational structure and workings as well as

to highlight the scientific accomplishments of NDACC over the past 25 years, before presenting further developments and perspective. This paper is definitely of interest for the community of data providers, cooperating networks and the broad community of users. No doubt that such an overview paper deserves publication. I recommend publication of this manuscript after that some points are considered.

It is worth noting that this paper also aims at being the introductory paper for the special issue "Twenty-five years of operations of the Network for the Detection of Atmospheric Composition Change (NDACC)". This way, the reader can expect to find here information on the other papers of the special issue or at least some highlights and context, but this is not exactly the case. This is actually my major concern with this manuscript. It misses a lot of references from the special issue and also from the past literature on topics where NDACC data set was definitely a major contributor. I give below four examples:

- Sub-section 3.1 "High-quality ozone datasets" does not mention the Steinbrecht et al., 2017 analysis (Atmos. Chem. Phys., 17, 10675-10690, https://doi.org/10.5194/acp-17-10675-2017) which is (i) part of the special issue and (ii) present the most recent results based on 14 NDACC stations measuring ozone profiles with different techniques.

- Sub-section 3.2 "Reference measurements for satellite validation" does not present any reference from the special issue. There is at least Zhou et al., 2016 (Atmos. Meas. Tech., 9, 5621-5636, https://doi.org/10.5194/amt-9-5621-2016), Khaykin et al., 2017 (Atmos. Chem. Phys., 17, 1829-1845, https://doi.org/10.5194/acp-17-1829-2017), Knepp et al., 2017 (Atmos. Meas. Tech. Discuss., https://doi.org/10.5194/amt-2017-90), etc. . .

- Sub-section 3.4 "Providing precise documentation of the multi-decadal trends of many tropospheric and stratospheric constituents" : At the end of the paragraph, Figure 12 and the reference Franco et al., 2016 illustrate the question of the Northern Hemisphere distribution and increase of ethane linked to oil and natural gas extraction. I think this

part of the manuscript should also refer to recent analysis by Helmig et al., Nature Geoscience 2016 (see complete reference below). Besides, The NDACC special issue is also containing a paper on this subject by Haussman et al., 2016 (Atmos. Chem. Phys., 16, 3227-3244, https://doi.org/10.5194/acp-16-3227-2016, 2016). I suggest to refer to this companion paper in the present manuscript.

- Sub-section 3.7 "Evaluating coupled-chemistry-climate models" presents no reference at all, neither from the special issue nor from the past literature, although NDACC data have been extensively used in the CCMVal activities for example. At least a few key reference should be given in this paragraph. In the special issue, there is also at least one paper related to this activity not cited in this manuscript (Zeng et al., 2017, Atmos. Chem. Phys., 17, 10495-10513, https://doi.org/10.5194/acp-17-10495-2017)

Minor comments :

- page 4, line 14 : double dot after "Isssue".

- page 13, line 6 : Why not mentioning CAMS in this list ?

- page 13, line 13 : It is is ambiguous to start a sentence by "Figure 5 shows . . ." and finishing it by ". . . ozonesondes (not included in Fig. 5. . .)". This last part of the sentence should be removed. A better solution would be to propose another sentence to justify why ozonesondes are not included in this figure. Besides, readers may also miss information/highlights on the NDACC activities regarding these ozonesondes in such an overview paper.

- page 15, line 13 : Anton is misspelled.

- page 27, line 6 : Acronyms and reference (or web site) should be given for AGAGE, HATS, Earlinet and MPLnet.

- page 27, line 10 : I'm afraid "e-infrastructure needs" is not clear enough for the reader. Needs clarification.

[Figure]

Reference :

Helmig, D., S. Rossabi, J. Hueber, P. Tans, S. Montzka, K. Masarie, K. Thoning, C. Plass-Duelmer, A. Claude, L. Carpenter, A.C. Lewis, S. Punjabi, S. Reimann, M. Vollmer, R. Steinbrecher, J. Hannigan, L. Emmons, E. Mahieu, B. Franco, D. Smale, and A. Pozzer, Reversal of global atmospheric ethane and propane trends largely due to US Oil and natural gas production. Nature Geoscience. Nature Geoscience, 2016. 9: p. 490–495.

---

## Author Comment (AC1) · 29 Jan 2018

| Answers to referees |
|---|
| M. De Mazière, A. Thompson, M. Kurylo, on behalf of all co-authors |
| Dec. 15, 2017 |

*NOTE: when referring to line numbers, we refer to line numbers in the initial version of the manuscript, before the revision.*

**Answers to referee #1**

**General comments**

- ➢ We added two sentences in the abstract (1) to better identify which kind of measurements are considered in NDACC (essentially remote-sensing measurements) and (2) to highlight the distinction between NDACC and other networks (compliance with strict measurement and data protocols to ensure high and consistent quality of the data).
- ➢ We have added some more references to (historical) scientific peer-reviewed papers where feasible, but we have the feeling that this is not so relevant, since this paper is an introduction to the Special Issue that includes 38 other peer-reviewed scientific papers linked to NDACC, each of them again referring to previous relevant NDACC-related papers.

**Minor comments**

- *Abstract The mission of NDACC does not become clear to me when reading the abstract. Is it mainly based on instruments and measurements that measure primarily stratospheric composition, or is it also focused on tropospheric composition? It may be good to list the different instruments (FTIR, lidar, ozonesondes, DOAS, Brewers, Dobsons: : :) that are actually part of NDACC (maybe in L28) so that the reader (if interested in a particular one of these) is informed right away.*

- *Introduction Although the history of air quality research is being mentioned in the introduction, it remains unclear if any NDACC related measurements are contributing to studying this issue. I would state somewhere more explicit whether NDACC measurements concentrate primarily on the composition of the middle atmosphere (or not if a contribution to air quality observations is made).*

- ➢ We added in the abstract an explicit enumeration of the NDACC measurements techniques.
- ➢ Question about scope of NDACC. The referee is right that it is not clear to what extent NDACC is concerned with air quality measurements.
  NDACC started as NDSC, focusing on the stratospheric ozone layer. When NDSC became NDACC, because the measurement techniques evolved and additional environmental issues became prominent, NDSC/NDACC broadened its scope to monitoring the composition of the atmosphere, from the troposphere to the mesosphere, in order to investigate the links between climate change, air quality and atmospheric composition.

Therefore, we moved the explicit listing of NDACC objectives to the Introdcution and we explicitly added the link to air quality.

- *P2L10 ': …like carbon dioxide, methane: : :' do you refer to ground-based in-situ measurements here?*
  ➢ Yes, this is about ground-based measurements. We clarified it in the text.
- *P2L16 A reference to Molina and Rowland, 1974 is here also needed.*
  ➢ We added that reference
- *P2L17/19/22/23 More references are needed after all these statements. This overview paper otherwise misses a chance to improve its educational value.*
  ➢ We added several references to historical papers
- *P2L29 Please provide reference to this report or where it can be found: we added the web-link to a PDF version of this report.*
  ➢ done
- P3L24 Add reference.
  ➢ The text has been clarified
- *P3L27 Add 'US' after 'Colorado':*
  ➢ done
- *P4 L9 Please introduce when the name change came along, the mentioning of NDACC for the first time in the introduction is otherwise too abrupt.:*
  ➢ done, see answer to comment about NDACC's mission statement. We moved the NDACC mission statement forward, to the end of Section 1.1. As such, the name NDACC is introduced before it appears in Sect. 1.2, and we answer the referee's remark concerning P9L10-17.
- *P7L2 Wording is unclear. Are there other measurements not included in this figure that also belong to NDACC, or are all measurements included shown?*
  ➢ A few more species are measured in NDACC, like CHOCHO (glyoxal) by the DOAS-type instruments, HCFC-142b by the FTIR spectrometers, etc. The list of species expands with time as the measurement techniques, research interests, and species' abundances in the atmosphere evolve. We added a sentence in the text to clarify this, and we added a small box listing all the variables for which data are currently available in the NDACC database.
- *Figure 2 Do you show here affiliated stations only? It would be good to have the candidate stations included as well. I ask because I wonder whether there is really no candidate station in India? I also thought that there I one or even two NDACC stations in Africa (Kenia and South Africa)? If true, an update of this figure would be needed.*
  ➢ We prefer not to include candidate stations because this map would show a quickly changing situation. But we added a sentence in the text that highlights the existence of candidate stations, showing that the network is growing, and especially so in the presently poorly covered regions of the world.
- *P9L2-4 Please rephrase sentence, I don't quite understand what you are driving at here:*
  ➢ Done: we have rephrased the sentence as "This quality assurance lends considerable credence to the ground-based record which NDACC has contributed to all the quadrennial WMO Scientific Assessments of Ozone Depletion (1991 to present)".
- *P9L10-17 This mission statement of NDACC would seem more logical and appropriate at the very beginning of Section 2, or even already in the introduction.*

- ➢ We moved the mission statement to the end of Sect. 1.1. and at the same time updated it according to our present understanding within NDACC.
- *Figure 3 Caption, why is the figure only similar, not equivalent to that in Figure 2? Also, I assume that theme and working groups are likely to overlap and it would be nice to reflect this in this diagram.*
  - ➢ We have updated the NDACC station map and organisation chart taking into account the referee's remarks.
  - ➢ It is true that the instrument working groups and theme groups have some overlap. But it would be very difficult to show that in the diagram – doing that would make the diagram quite fuzzy. So we added a sentence in the manuscript to indicate such overlaps.  Also we moved the paragraph on the theme groups higher up in the section, before the discussion on the cooperating networks – where it fits more logically.
- *P11L1 This information sounds incomplete. Why do you only show three of the existing theme groups? It would be more valuable to provide the most up-to-date information even though this may mean that figure 3 would have to be updated.*
  - ➢ In fact, at the latest NDACC SC meeting in November 2017, we decided on  4 theme groups in total. So we updated the text and the chart
- *P13L1 and elsewhere:*
  - ➢ we agree with the referee that we should always use the same acronym/name – so we have adopted NDACC everywhere in the paper after Section 1.1. We inform the reader of this choice at the end of Section 1.1
- *P13L11 GOZCARDS should be listed in the list with SAGE/OSIRIS, SWOOSH, and ESA-CCI, since it is composed of the same kind of satellite instruments, not just after SBUV-MOD.*
  - ➢ done
- *P13L13 It seems inappropriate to highlight the ground-based data here as being 'highquality', since it implies the satellite measurements would not be high-quality. Please remove.*
  - ➢ Done. We replaced the word high-quality by reference.
- *P13L13 It is confusing to include ozonesondes first, but then right away exclude them again, since they are not in the figure. Simply describe what is in the figure!*
  - ➢ We removed the mention of ozonesondes.
- *P13L17 Title would be more meaningful with 'Monitoring long-term atmospheric composition change'*
  - ➢ We changed the title to 'Long-term ozone monitoring' – because this section is only about ozone, in contrast to sections 3.3 and 3.4 which are mainly about other stratospheric and tropospheric species.
- *P13L18-24 To my knowledge is it outdated to talk about the three stages of ozone recovery. If you want to stick to this, refer to the respective WMO ozone assessment in which these stages were discussed. Also, from more elaborate statistical evaluations than that presented here, it is not clear whether the second stage of ozone recovery is reached already. Please remove this statement or provide references as a proof.*
  - ➢ We rephrased that part of the text while removing any reference to the three stages of ozone recovery.
- *Figure 5 caption 'ESC' is 'effective stratospheric chlorine', not what the authors indicate here. You may also want to add a reference to where this figure first appeared in a publication, I assume updated from WMO 2014?*

- ➢ The referee is right that ESC stands for 'Effective stratospheric chlorine' and so the caption of Fig. 5 has been corrected for it.
- *P15L12/Figure 6 It's incorrect to say that figure 6 shows the limb satellite sounders. Please rewrite or produce a new figure.*
  - ➢ We have added a Fig. 6b that shows the limb and occultation sounders
- *P17L10 "NDACC excels: : :" is this not a little bit an overstatement? It may be true for any trace gas the lidars or ozonesondes may measure, but certainly for minor trace gas species, the resolution of the ACE-FTS satellite limb sounder (approx. 1-3 km in the UTLS) is not achieved?*
  - ➢ The referee is right that this is an overstatement. We have rewritten the statement, pointing specifically to the NDACC instruments for which the statement is appropriate, and removing the part about the vertical resolution of the satellite instruments in the UT/LS, as follows: "NDACC lidar and sonde instruments provide insights into the upper troposphere/lower stratosphere (UT/LS) where most satellite limb measurements are less precise than in the middle and upper stratosphere. "
- *P18L6 Please improve the English of this sentence.*
  - ➢ Done
- *Section heading 3.3 Suggest shortening to 'Constraining uncertainties in ozone absorption*
- *cross-sections'*
  - ➢ Done.
- *Section 3.4 This section would be more appropriate/logical to move to right after (or included within) section 3.1. See suggestion of header change above.*
  - ➢ After reflection, we have moved Section 3.3 to right after Section 3.1 because both talk about ozone. Section 3.4 on the other hand talks about long-term trend evaluations, not only of ozone but about several tropospheric and stratospheric constituents, which fits well, to our opinion, after the original Section 3.2 about the support to satellite validation.
- *Figure 11 Are AGAGE and HATS measurements really part of the NDACC system? If not, I would not tend to include this figure and discussion in the achievements section here. If yes, since you label it here EESC (in contrast to ESC in an earlier figure) it would be appropriate to explain to the general reader how these variables differ from each other.*
  - ➢ We have removed Fig. 11 and adapted the text accordingly.
- *Figure caption 12 I do not understand the information given in this figure caption and the main result that this figure represents. Please rework and be more specific also in the text.*
  - ➢ We have rewritten the caption and added the conclusion that has been drawn from this work in the text.
- *P23L8 To have a reference on stratospheric water vapour changes from continuous ground-based measurements will become particularly important in the future and for future climate considerations, since past trends derived from merged limb satellite datasets have been yielding controversial results (cf. Hegglin et al., Nature Geoscience 2014).*
  - ➢ This is a useful remark by the referee and we added a sentence in that sense in the text.
- *P23L20 This definition sounds unnecessarily complicated to me, please consider rewriting.*
  - ➢ we simplified it to: 'the UV dose on a horizontal surface causing sunburn'
- *Section 3.6 The title of this section is more complicated than necessary. Why do you say 'bounding factors', are these not direct UV measurements?*
  - ➢ We changed it to: 'Latitudinal differences of UV-A and erythemal ("sunburning") radiation'

- *P25L23 What constituents have been evaluated in this context within CCMVal and in which papers? Please expand and provide references.*
  - ➢ We have added material to the first paragraph of Section 3.7 stating that HCl and ClONO2 column data sets were used in the CCMVal report and in a follow-on study. We specify that the NDACC measurements were used to identify problems in simulated HCl/ClONO2 partitioning and that led to the recognition of how transport in models directly affects ozone chemistry and projections of future ozone.
- *Section 4.2 I have to admit that I was somewhat lost reading the description of this tiered system of systems concept by Thorne et al (2017). I assume a score of 1 is the worst mark, a score of 6 the best mark achievable? What are the requirements to achieve the label of a global reference network?*
  - ➢ The network maturity is evaluated by assigning scores to several characteristics of the network dealing with measurement traceability, standard operation and processing procedures, documentation, data availability, sustainability etc.  Maturity scores 5 and 6 establish a reference measurement capability. The more network characteristics have a score 5 or 6, the more the network approaches a true reference network.
    Since the first submission of this manuscript, the paper by Thorne et al. (2017) has been revised and published in its final form, and hopefully, it is more intelligible now.
- *P28L14 What was the reasoning for using HDF? Wouldn't it have been easier for the community to use NetCDF as common format as mostly used in the climate modeling and Obs4MIP as well?*
  - ➢ The HDF format was chosen around the year 2000, in agreement with the satellite community that used the HDF format at that time. At that time, NetCDF was not yet adopted commonly by the climate modelling community. This being said, the NDACC DHF will provide a tool that enables the user to convert from the GEOMS HDF to the NetCDF format. This has been added in the text.
- *P30L16 suggest to add 'for validation', or did you refer to something else here?*
  - ➢ We added 'for validation' as this was what we referred to.
- *P31L1 In fact, the lack of plans to fly limb satellite sounders operationally highlights the urgent need to maintain a strong ground-based measurement system so to be able to support the WMO in its task to assess the state of the ozone layer every 4 years as requested by the Montreal Protocol.*
  - ➢ This is a useful remark by the referee: we added it in the text.
- *P31L25 Add reference to Morgenstern et al. (ACP, 2017)*
  - ➢ Done

*Technical corrections*

- *P4L24 Check punctuation.*
- *P8L1 put comma after 'France'*
- *P13L19 Missing bracket*
- *Figure 9 caption change 'over the stations' to 'above the stations'*
- *Figure 10 caption correct 'chorine' to 'chlorine'*
- *Figure 12 caption correct 'Norther' to 'Northern'*
- *Figure 14 caption remove extra space after 'Barrow'*
  - ➢ All have been corrected.

---

## Author Comment (AC2) · 29 Jan 2018

The comment was uploaded in the form of a supplement: https://www.atmos-chem-phys-discuss.net/acp-2017-402/acp-2017-402-AC2-supplement.pdf

---

## Author Response (AR2)

**Answer to co-editor comments:**

- p.1 (abstract): "all measurement in the Network are performed by *ground-based* remote sensing techniques" (just to be clear for a general readership).

Corrected

- p.13: "Copernicus Atmosphere Monitoring Service" (not "Atmospheric").

10   Corrected

- p.31-32 (section 4.2): in spite of the revision and reference to (Thorne et al., 2017), I am afraid that the section remains somewhat obscure because it is either too short or too long. Figure 15 is especially very hard to understand fully: the meaning of lines in the matrix is unclear and the terms "comprehensive", "baseline" (etc...) are not really used in the

15   plain English sense. In the end, I suggest either to add a few paragraphs in order to explain better the maturity matrix or to remove Figure 15 and make the same point (NDACC is a mature network) more qualitatively and refer to (Thorne et al., 2017) for details.

The section has been rewritten and the Figure removed.

- some figures are low quality, please revise. This is not only an issue with e.g. the DPI output, but mainly with the fact that some graphics (especially Figs 1 and 3) are originally slides designed for projection on a large screen. In the paper, the use of tiny fonts and of many colours does not work too well. Could you try revise/simplify some of the Figures?

25   Some figures have been revised.
Figures 1 and 3 have been designed not only for projection on a large screen, and their original format allows zooming in without any problem. The problem that has arisen with several figures is that their embedding in the word file degraded the quality. Since we send all figures now separately in the supplemental .zip file, it will be clear that the quality of Figures 1 and 3 (and the other ones) is very good and appropriate for ACP/AMT/ESSD publication.

-   In the paragraph that goes with Fig. 5, minor editorial changes have been made.

-   Some names that were forgotten have been added in the acknowledgements; the list is now alphabetically ordered.

[revised manuscript text omitted]